# CyAbrB2 is a nucleoid-associated protein in *Synechocystis* controlling hydrogenase expression during fermentation

Ryo Kariyazono, Takashi Osanai*

School of Agriculture, Meiji University, Kawasaki, Japan

**Abstract** The *hox* operon in *Synechocystis* sp. PCC 6803, encoding bidirectional hydrogenase responsible for H$_2$ production, is transcriptionally upregulated under microoxic conditions. Although several regulators for *hox* transcription have been identified, their dynamics and higher-order DNA structure of *hox* region in microoxic conditions remain elusive. We focused on key regulators for the *hox* operon: cyAbrB2, a conserved regulator in cyanobacteria, and SigE, an alternative sigma factor. Chromatin immunoprecipitation sequencing revealed that cyAbrB2 binds to the *hox* promoter region under aerobic conditions, with its binding being flattened in microoxic conditions. Concurrently, SigE exhibited increased localization to the *hox* promoter under microoxic conditions. Genome-wide analysis revealed that cyAbrB2 binds broadly to AT-rich genome regions and represses gene expression. Moreover, we demonstrated the physical interactions of the *hox* promoter region with its distal genomic loci. Both the transition to microoxic conditions and the absence of cyAbrB2 influenced the chromosomal interaction. From these results, we propose that cyAbrB2 is a cyanobacterial nucleoid-associated protein (NAP), modulating chromosomal conformation, which blocks RNA polymerase from the *hox* promoter in aerobic conditions. We further infer that cyAbrB2, with altered localization pattern upon microoxic conditions, modifies chromosomal conformation in microoxic conditions, which allows SigE-containing RNA polymerase to access the *hox* promoter. The coordinated actions of this NAP and the alternative sigma factor are crucial for the proper *hox* expression in microoxic conditions. Our results highlight the impact of cyanobacterial chromosome conformation and NAPs on transcription, which have been insufficiently investigated.

## eLife assessment

The authors provide **solid** data on a functional investigation of potential nucleoid-associated proteins and the modulation of chromosomal conformation in a model cyanobacterium. These **valuable** findings will be of interest to the chromosome and microbiology fields. Additional analysis and the tempering of conclusions has helped to improve the work, although further refinement remains possible.

**\*For correspondence:**
tosanai@meiji.ac.jp

**Competing interest:** The authors declare that no competing interests exist.

## Introduction

Cyanobacteria perform fermentation, using glycolytic products as electron acceptors (*Stal and Moezelaar, 1997*). Cyanobacteria have multiple fermentation pathways according to the environment. For example, the freshwater living cyanobacterium *Synechocystis* sp. PCC 6803 (hereafter referred to as *Synechocystis*) produces acetate, lactate, dicarboxylic acids, and hydrogen (*Stal and Moezelaar, 1997*; *Osanai et al., 2015*).

Hydrogen is generated in quantities comparable to lactate and dicarboxylic acids as the result of electron acceptance in the dark microoxic condition (*Iijima et al., 2016*; *Akiyama and Osanai, 2023*).

Bidirectional hydrogenase is a key enzyme for $H_2$ production from protons (*Carrieri et al., 2011*) and is commonly found in cyanobacteria (*Puggioni et al., 2016*). Cyanobacterial hydrogenase comprises five subunits (HoxEFUHY) containing nickel and Fe-S clusters (*Cassier-Chauvat et al., 2014*). This enzyme can utilize NADH, reduced ferredoxin, and flavodoxin as substrates (*Gutekunst et al., 2014*). Hydrogenase mainly receives reduced ferredoxin from pyruvate-ferredoxin oxidoreductase (PFOR) in the microoxic condition (*Gutekunst et al., 2014*; *Artz et al., 2020*).

Although hydrogenase and PFOR are $O_2$ sensitive, they can work under aerobic conditions (*Wang et al., 2021*; *Burgstaller et al., 2022*; *Appel et al., 2000*). Therefore, uncontrolled expression of *hox* operon and *nifJ* (coding gene of PFOR) may hamper metabolism under photosynthetic conditions. Furthermore, genetic manipulations on *Synechocystis* have demonstrated that modulating the expression of certain enzymes including hydrogenase can alter fermentative metabolic flow (*Iijima et al., 2016*; *Akiyama and Osanai, 2023*; *Iijima et al., 2021*). This provides evidence that transcription regulates the fermentative pathway. Thus, transcriptional regulation in response to the environment is essential for optimal energy cost performance.

Promoter recognition by RNA polymerases is an essential step in transcriptional regulation. Sigma factors, subunits of RNA polymerase, recognize core promoter sequences. Transcription factors can also bind to promoter regions to suppress or promote RNA polymerase transcription. As well as recruitment or blocking of RNA polymerase, some transcriptional factors, known as nucleoid-associated proteins (NAPs), modulate chromosomal conformation to regulate transcription (*Hołówka and Zakrzewska-Czerwińska, 2020*). NAPs are common in bacteria, but cyanobacterial NAPs remain unidentified, and higher-order DNA structure in cyanobacteria is yet to be shown. A recent study suggested that the manipulation of chromosomal supercoiling impacts transcriptional properties in cyanobacteria (*Behle et al., 2022*). There is room for consideration of NAPs modulating chromosomal conformation and regulating expression in cyanobacteria.

In *Synechosysits*, the coding genes of HoxEFUHY form a single operon (sll1220–1226), while PFOR is encoded in the *nifJ* (sll0741) gene. Both *hox* and *nifJ* operons are highly expressed under microoxic conditions (*Summerfield et al., 2011*). Genetic analysis has revealed that multiple global transcriptional regulators control *hox* and *nifJ* expression. Sigma factor SigE (Sll1689) promotes the expression of *hox* and *nifJ* operons (*Osanai et al., 2005*; *Osanai et al., 2011*), while transcription factor cyAbrB2 (Sll0822) represses them (*Dutheil et al., 2012*; *Leplat et al., 2013*). Positive regulators for the *hox* operon include LexA (Sll1626) and cyAbrB1 (Sll0359) (*Oliveira and Lindblad, 2008*; *Gutekunst et al., 2005*; *Oliveira and Lindblad, 2005*).

SigE, an alternative sigma factor, controls the expression of genes involved in glycogen catabolism and glycolysis, as well as PFOR/*nifJ* and hydrogenase (*Osanai et al., 2005*). SigE shows a high amino acid sequence similarity with the primary sigma factor SigA, which is responsible for transcribing housekeeping and photosynthetic genes (*Imamura and Asayama, 2009*). A ChIP-seq study revealed that, while most SigE binding sites are the same as SigA, SigE exclusively occupies the promoters of glycogen catabolism and glycolysis (*Kariyazono and Osanai, 2022*).

CyAbrB2 and its homolog cyAbrB1 are transcription factors highly conserved in cyanobacteria. For example, cyAbrB homologs in *Anabaena* sp. PCC7120 is involved in heterocyst formation (*Higo et al., 2019*). CyAbrB2 in *Synechocystis* regulates the expression of several genes involved in carbon metabolism, nitrogen metabolism, and cell division (*Leplat et al., 2013*; *Ishii and Hihara, 2008*; *Lieman-Hurwitz et al., 2009*). CyAbrB2 binds to the *hox* promoter in vitro and represses its expression in vivo (*Dutheil et al., 2012*). CyAbrB1, an essential gene, physically interacts with the cyAbrB2 protein (*Yamauchi et al., 2011*) and binds the *hox* promotor region in vitro to promote its expression (*Oliveira and Lindblad, 2008*).

To explore the dynamics of those transcription factors governing the expression of *hox* operon, we conducted a time-course analysis of the transcriptome and ChIP-seq of SigE and cyAbrB2. Our ChIP-seq and transcriptome analysis showed the NAPs-like nature of cyAbrB2, which prompted us to conduct a chromosomal conformation capture assay. 3C analysis explored the physical interaction between the *hox* promoter region and its downstream and upstream genomic region in the aerobic condition, and some loci changed interaction frequency upon entry to the microoxic condition. Furthermore, some interactions in the Δ*cyabrB2* mutant were different from those of the wild-type. From those experiments, we propose that cyAbrB2 modulates chromosomal conformation like NAPs, allowing access to the SigE-containing RNA polymerase complex on the *hox* promoter, by

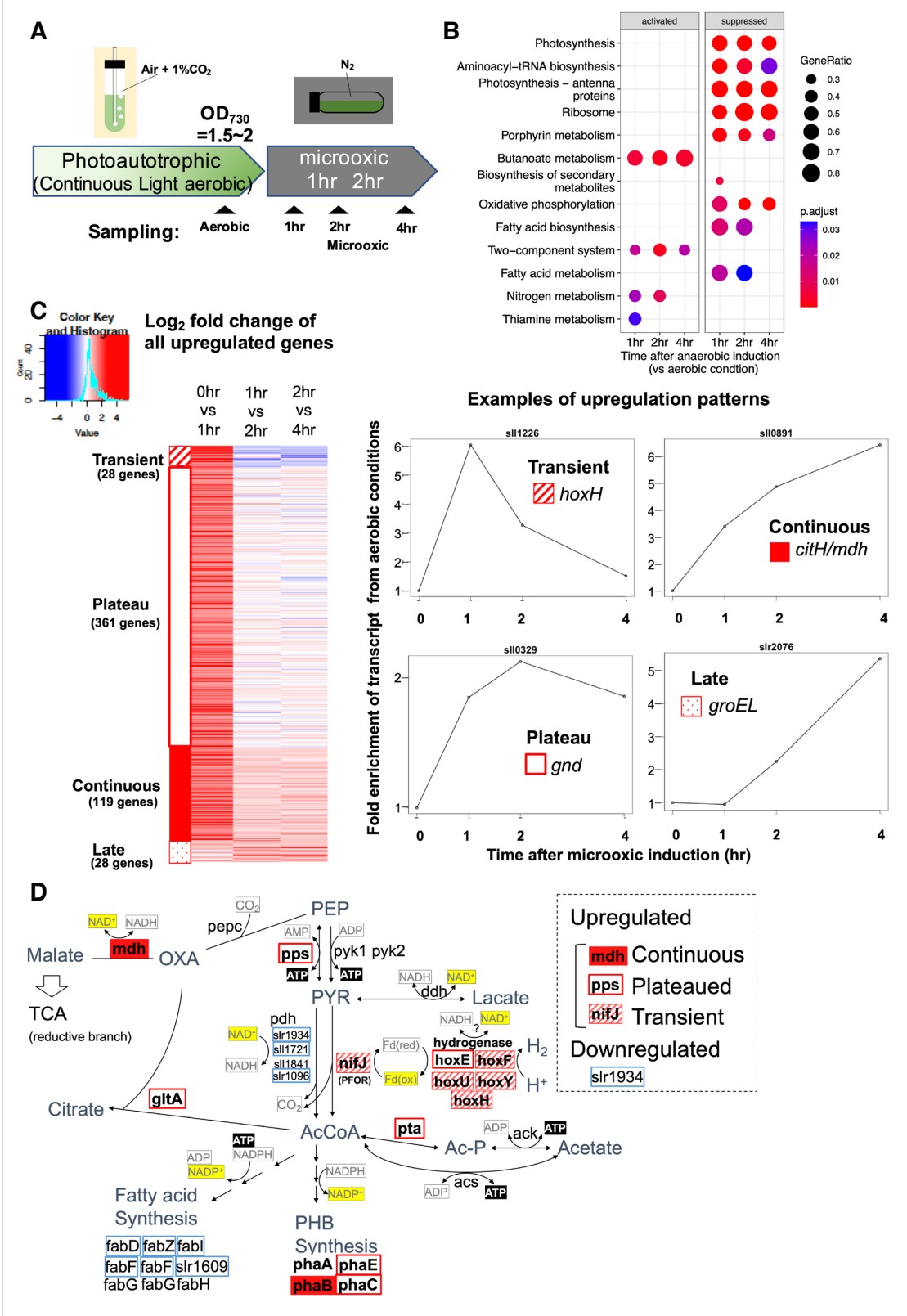

**Figure 1.** Time-course analysis of the transcriptome of *Synechocystis* on entry to the microoxic conditions. (**A**) Schematic diagram for the sampling of cells under aerobic and microoxic conditions. (**B**) Gene set enrichment analysis on time-course transcriptome data. KEGG pathways enriched in upregulated or downregulated genes after 1, 2, and 4 hr incubation under microoxic conditions are shown. (**C**) (Left) Heatmap showing expression change in all upregulated genes over the time course. Genes classified into transient (striped square), plateau (open square), continuous (filled square),

*Figure 1 continued on next page*

*Figure 1 continued*

and late (dotty square) were clustered into subgroups and sorted by the gene name. (Right) Examples of genes are classified into each expression pattern. (**D**) The classified genes were mapped to central carbon metabolism, centered on pyruvate. PEP: phosphoenolpyruvate, PYR: pyruvate, AcCoA: acetyl CoA, Ac-P: acetyl phosphate, OXA: oxaloacetate, PHB: polyhydroxy butyrate, TCA: tricarboxylic acid cycle.

The online version of this article includes the following figure supplement(s) for figure 1:

**Figure supplement 1.** Schematic diagram showing the classification of genes according to the time-course transcriptome.

which the *hox* operon achieves distinct expression dynamics. Chromosomal conformation of bacteria is a growing area of interest, and the findings of this study have brought insight into the transcriptional regulation of cyanobacteria.

## Results

### Transcriptomes on entry to dark microoxic conditions

To understand transcriptional regulation under microoxic conditions, we conducted a time-course transcriptome capturing light aerobic and dark microoxic conditions at 1, 2, and 4 hr timepoints (*Figure 1A*). Gene set enrichment analysis (GSEA) based on KEGG pathway revealed that many biological pathways, including photosynthesis and respiration (oxidative phosphorylation), were downregulated by the transition to dark microoxic conditions from light aerobic conditions (*Figure 1B*). Upregulated pathways included butanoate metabolism and two-component systems. The enrichment in the butanoate metabolism pathway indicates the upregulation of genes involved in carbohydrate metabolism. We further classified genes according to their expression dynamics. Within 1 hr of switching from aerobic to microoxic conditions, the expression levels of 508 genes increased more than twofold. Furthermore, genes with increased expression levels were classified into four groups based on the time course (*Figure 1C* and *Figure 1—figure supplement 1*). Of the 508 genes, 28 were termed 'transiently upregulated genes' due to their decreased expression upon the comparison of 1 and 4 hr incubation under microoxic conditions (Log2 fold change < −0.5), and 119 were termed 'continuously upregulated genes', which continuously increased between 1 and 4 hr incubation under microoxic conditions (Log2 fold change >0.5). Other than 508 genes twofold upregulated within 1 hr, 28 genes showed more than fourfold increment within 4 hr but not upregulated within 1 hr. We combined those 'Late upregulated genes' with 508 genes and referred to as 'All upregulated genes' in the subsequent analysis (*Figure 1—figure supplement 1*). Mapping the classified genes to central carbon metabolism revealed that *nifJ* encoding PFOR and *hox* operon encoding a bidirectional hydrogenase complex were transiently upregulated (*Figure 1D* and *Table 1*). RT-qPCR verified the transient expression of *hoxF*, *hoxH*, and *nifJ* (*Figure 2—figure supplement 1*).

### SigE and cyAbrB2 control the expression of transiently upregulated genes

The functional correlation between hydrogenase and PFOR, encoded by the *hox* operon and *nifJ*, suggests that transient upregulation has physiological significance. We focused on transiently upregulated genes and attempted to reveal the regulatory mechanism underlying transient upregulation. While SigE promotes the expression of *hox* and *nifJ*, cyAbrB2 represses them (*Osanai et al., 2005*; *Dutheil et al., 2012*; *Leplat et al., 2013*). We confirmed that the deletion of *sigE* and *cyabrb2* (Δ*sigE* and Δ*cyabrb2*, respectively) affected the expression of *hoxF*, *hoxH*, and *nifJ* by RT-qPCR (*Figure 2—figure supplement 1*). Thus, we conducted a time-course transcriptome analysis of Δ*sigE* and Δ*cyabrb2* under aerobic conditions and after 1 and 2 hr cultivation in dark microoxic conditions (*Figure 2A* and *Figure 2—figure supplement 2*). The transcriptome data confirmed that SigE and cyAbrB2 regulated *hox* operon expression (*Figure 2B*). At each timepoint, we searched for differentially expressed genes (DEGs) between mutants and wildtype with a more than twofold expression change and a false discovery rate (FDR) <0.05. We found that deleting *sigE* or *cyabrb2* preferentially affected the expression of transiently upregulated genes, not limited to *hox* and *nifJ* operons (*Figure 2C and D*). Interestingly, *cyabrb2* deletion resulted in the upregulated expression of transient genes under aerobic conditions, in contrast to 1 hr cultivation under microoxic conditions (*Figure 2C*).

**Table 1.** List of transiently upregulated genes.

| | | Operon |
|---|---|---|
| **Oxidoreductase** | | |
| sll0741 | *nifJ*/'pyruvate-ferredoxin/flavodoxin oxidoreductase' | |
| sll0743 | Hypothetical protein | |
| sll0744 | Dihydroorotate dehydrogenase (fumarate) | TU3296 |
| sll1221 | *hoxF*/'bidirectional [NiFe] hydrogenase diaphorase subunit' | |
| sll1222 | Unknown protein | |
| sll1223 | *hoxU*/'bidirectional [NiFe] hydrogenase diaphorase subunit' | |
| sll1224 | *hoxY*/'NAD-reducing hydrogenase small subunit' | |
| sll1225 | Unknown protein | |
| sll1226 | *hoxH*/'NAD-reducing hydrogenase large subunit' | TU1714 |
| slr1434 | *pntB*/'H+-translocating NAD(P) transhydrogenase subunit beta' | TU1089 |
| **Transporter** | | |
| sll1450 | *nrtA*/'nitrate/nitrite transport system substrate binding protein' | |
| sll1451 | *nrtB*/'nitrate/nitrite transport system permease protein' | |
| sll1452 | *nrtC*/'nitrate/nitrite transport system ATP binding protein' | |
| sll1453 | *nrtD*/'nitrate/nitrite transport system ATP binding protein' | TU1023 |
| **Two-component system** | | |
| slr1214 | Twitching motility two-component system response regulator PilG | TU905 |
| slr1215 | Unknown protein | TU907 |
| **Glycosyl transferase** | | |
| slr2116 | *spsA*/'spore coat polysaccharide biosynthesis protein; SpsA' | TU1673 |
| **Protease** | | |
| sll1009 | *frpC*/'iron-regulated protein' | TU491 |
| **Insertion sequence (transposase)** | | |
| slr1523 | Transposase | TU1659 |
| sll1985 | Transposase | TU1589 |
| sll7001 | Transposase | NA |
| sll7003 | Toxin FitB | TU7001 |
| ssl0172 | Transposase | TU3163 |
| **Other** | | |
| slr1260 | Hypothetical protein | TU1446 |
| slr0668 | Unknown protein | TU3532 |
| slr5127 | Unknown protein | TU5127 |
| sll0710 | Unknown protein | TU97 |
| sll1307 | Unknown protein | TU1224 |

The list of transiently upregulated genes was merged by transcriptional units and sorted by function. The transcriptional unit information was obtained from a previous study (**Kopf et al., 2014**).

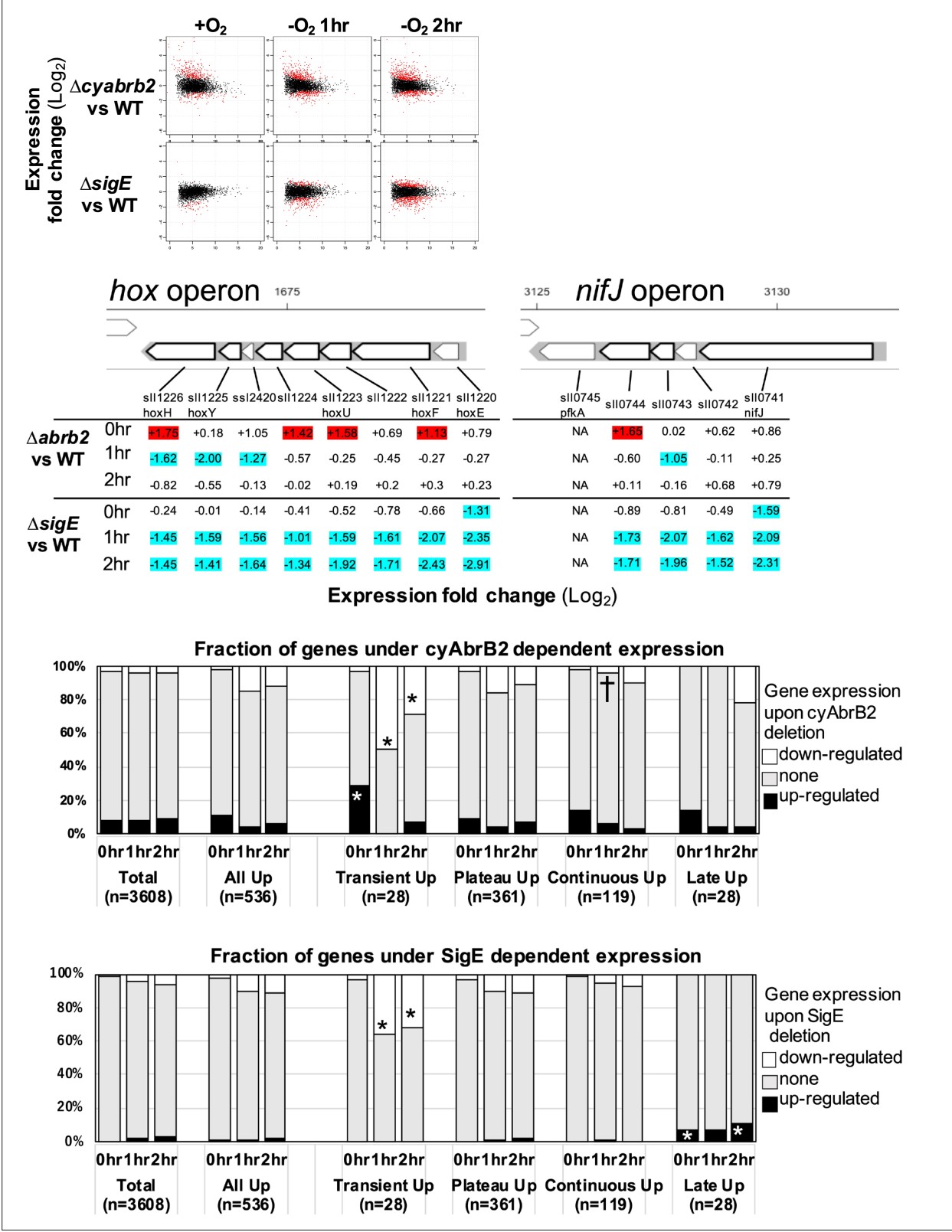

**Figure 2.** The impacts of Δ*sigE* and Δ*cyabrb2* on the time-course transcriptome. (**A**) MA plot showing fold change (y-axis) and average (x-axis) of gene expression between wildtype and mutant strains at each timepoint. Red dots indicate defined differentially expressed genes (DEGs) (|Log$_2$ FC|>1 with false discovery rate [FDR]<0.05). (**B**) Log2 scaled expression fold change in genes in the *hox* and *nifJ* operons upon Δ*cyabrb2* and Δ*sigE* under aerobic conditions (0 hr), 1 hr after microoxic condition (1 hr), and 2 hr after microoxic condition (2 hr). DEGs are marked with sky blue (downregulated upon

*Figure 2 continued on next page*

*Figure 2 continued*

deletion) or red (upregulated upon deletion). (**C and D**) Fraction of upregulated and downregulated genes upon the (**C**) Δ*cyabrb2* and (**D**) Δ*sigE* at the timepoints of aerobic conditions (0 hr), 1 hr after anoxic condition (1 hr), and 2 hr after anoxic condition (2 hr). Genes are classified according to *Figure 1C*. Asterisk (*) and dagger (†) denote statistically significant enrichment and anti-enrichment compared with all upregulated genes tested by multiple comparisons of Fisher's exact test (FDR<0.05).

The online version of this article includes the following figure supplement(s) for figure 2:

**Figure supplement 1.** RT-qPCR validated the transiently upregulated genes classified by RNA-seq.

**Figure supplement 2.** Primary component scatter plot showing the profiles of RNA-seq data.

## Genome-wide analysis of cyAbrB2, cyAbrB1, and SigE via ChIP-seq

To decipher the regulatory mechanism of transiently upregulated genes, we must first comprehend the fundamental features and functions of these transcriptional regulators. Therefore, a genome-wide survey of cyAbrB2 and SigE occupation (*Figure 3—figure supplement 1*) combined with transcriptome data was done. Specifically, we generated a *Synechocystis* strain in which cyAbrB2 was epitope-tagged and performed a ChIP-seq assay under aerobic and microoxic conditions (*Figure 3—figure supplements 2 and 3*). SigE-tagged strains previously constructed and analyzed elsewhere were also employed (*Kariyazono and Osanai, 2022*). The primary sigma factor SigA was also analyzed to determine SigE-specific binding. In addition to cyAbrB2, we tagged and analyzed cyAbrB1, which is the interactor of cyAbrB2 and positively regulates the *hox* operon.

## CyAbrB2 binds to long tracts of the genomic region and suppresses genes in the binding region

The ChIP-seq data showed that cyAbrB2 bound to long tracts of the genomic region with lower GC content than the whole-genome *Synechocystis* (*Figure 3A and B*). Vice versa, regions exhibiting lower GC contents showed a greater binding signal of cyAbrB2 (*Figure 3C*). This correlation was not a systematic bias of next-generation sequencing because the binding signals of SigE, SigA, and control showed no negative correlation to GC contents (*Figure 3—figure supplement 4*). The binding regions of cyAbrB2 called by peak caller covered 15.7% of the entire genome length. 805 of 3614 genes overlapped with cyAbrB2 binding regions, and almost half (399 of 805 genes) were entirely covered by cyAbrB2 binding regions. The cyAbrB2 binding regions included 80 of 125 insertion sequence elements (*Figure 3D*). Comparison with the transcriptome of Δ*cyabrB2* revealed that cyAbrB2 tended to suppress the genes overlapping with its binding regions under aerobic conditions (*Figure 3A and E*). A survey of the genomic localization of cyAbrB1 under aerobic conditions revealed that cyAbrB1 and cyAbrB2 shared similar binding patterns (*Figure 3A* and *Figure 3—figure supplement 5A*). Due to the essentiality of cyAbrB1, we did not perform transcriptome analysis on a cyAbrB1-disrupted strain. Instead, we referred to the recent study performing transcriptome of cyAbrB1 knockdown strain, whose cyAbrB1 protein amount drops by half (*Hishida et al., 2024*). Among 24 genes induced by cyAbrB1 knockdown, 12 genes are differentially downregulated genes in *cyabrb2*Δ in our study (*Figure 3—figure supplement 5*).

## CyAbrB2 binds to transiently upregulated genes

The binding regions of cyAbrB2 overlapped 17 of 28 transiently upregulated genes, showing significant enrichment from all upregulated genes (*Figure 4A*). The transiently upregulated genes belong to 17 transcriptional units (TUs), according to the previous study (*Kopf et al., 2014*), and cyAbrB2 tends to bind TUs with transiently upregulated genes (*Figure 4B*). While cyAbrB2 covered the entire length of insertion sequences and unknown proteins, its binding positions on other transient genes were diverse (*Figure 4C*). Specifically, the *hox* and *nifJ* operons had two distinct binding regions located at the transcription start sites (TSSs) and middle of operons, the *pntAB* operon had two binding regions in the middle and downstream of the operon, and the *nrtABCD* operon had one binding region downstream of the operon (*Figure 4C*).

## Localization of cyAbrB2 became blurry under the microoxic condition

When cells entered microoxic conditions, the relative ChIP-seq signals in the cyAbrB2 binding regions slightly declined (*Figure 5A and B*). Notably, the total quantities of precipitated DNA by tagged

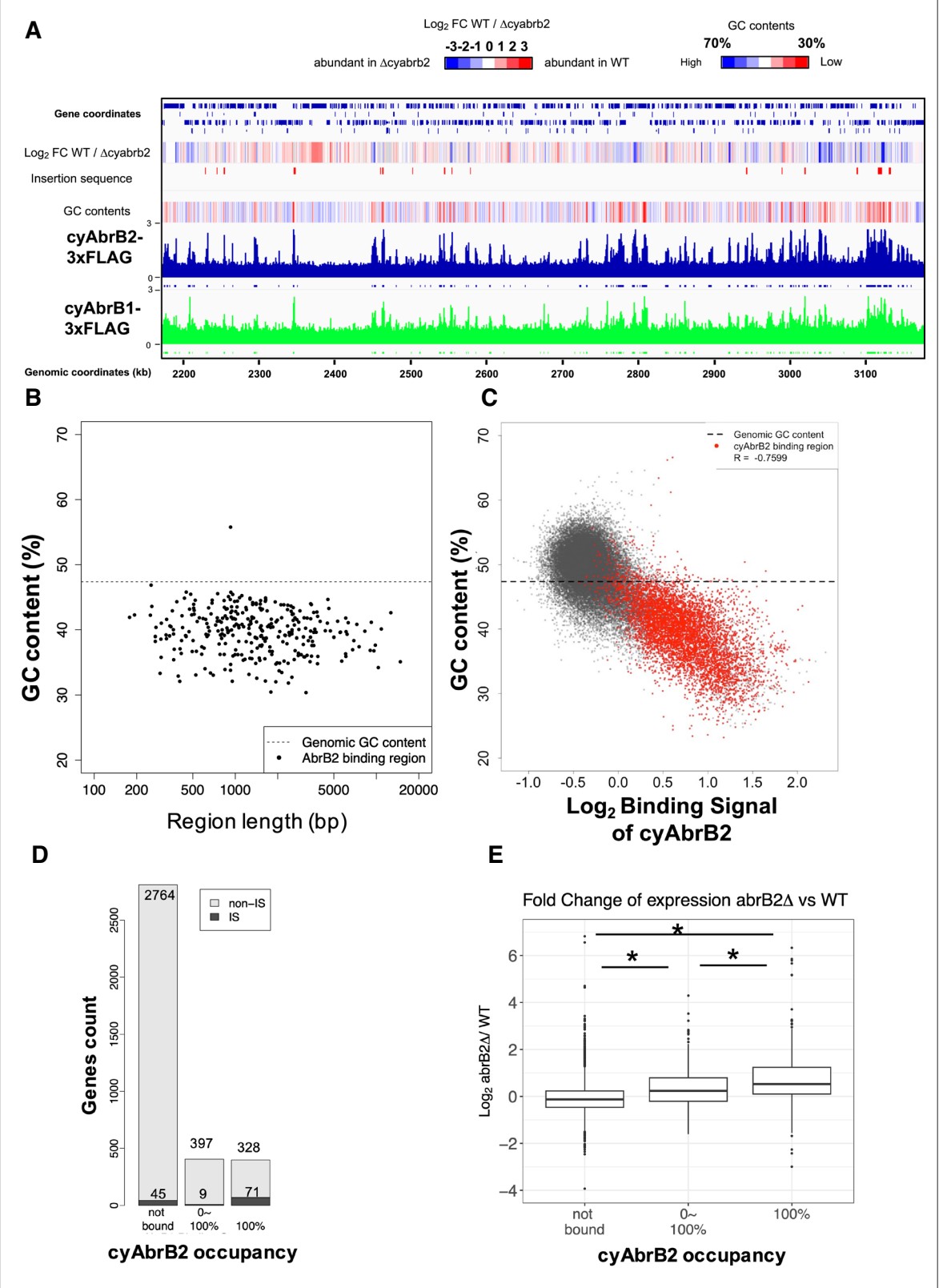

**Figure 3.** The long-tract distribution of cyAbrB2 on the *Synechocystis* genome and the repressive effect of cyAbrB2 on the gene expression.
(**A**) Snapshot of ChIP-seq data for cyAbrB2 and cyAbrB1 under aerobic conditions. The heatmap in the second column indicates expression fold change upon Δ*cyabrb2* under aerobic conditions. Positive values (colored in red) indicate that the gene expression is higher in wildtype than in Δ*cyabrb2*, and negative values (colored in blue) indicate the opposite. The positions for the insertion elements are marked with red in the third column. The heatmap

*Figure 3 continued on next page*

*Figure 3 continued*

in the fourth column indicates GC contents. High GC contents are colored in blue and low GC contents are colored in blue. (**B**) GC contents and region length of cyAbrB2 binding regions (black dots). The horizontal dotted line indicates the genomic average of GC content. (**C**) Scatter plot of GC content and binding signal of cyAbrB2. The x-axis is the binding signal of cyAbrB2 in each 100 bp region, and the y-axis indicates GC contents within 500 bp windows sliding every 100 base pairs. CyAbrB2 binding regions are marked with red dots. (**D**) Histogram of genes showing the extent of occupancy (not bound, partially overlapped, or entirely overlapped) by the cyAbrB2 binding region. The gray bars indicate non-IS genes, and the count numbers of the non-IS genes are displayed on the gray bars. The black bars indicate the IS genes, and the count numbers of the IS genes are displayed above the black bars. (**E**) Boxplot showing fold change in gene expression by Δ*cyabrb2* under aerobic conditions. Genes are classified according to the extent of occupancy by the cyAbrB2 binding region. Asterisk (*) denotes statistical significance tested by multiple comparisons of the Wilcoxon-rank test. Actual FDRs of "not bound vs 0~100%", "not bound vs 100%", and "0~100% vs 100%" are <2e-16, <2e-16, and 5e-06, respectively.

The online version of this article includes the following figure supplement(s) for figure 3:

**Figure supplement 1.** Overview of genome occupancy of cyAbrB2, cyAbrB1, SigE and SigA under the aerobic and microoxic conditions.

**Figure supplement 2.** Validation of procedure for ChIP-seq of FLAG-tagged cyAbrB2, SigE, and SigA.

**Figure supplement 3.** Confirmation of genomic deletion and the epitope tagging of *abrB2* (#1-#3), the epitope tagging of *abrB1* (#4 and #5), and deletion of *sigE* (#6 and 7).

**Figure supplement 4.** Relationships between GC content and binding patterns for SigE and SigA.

**Figure supplement 5.** cyAbrB2 and cyAbrB1 show similar binding pattern and overlapping gene regulation.

cyAbrB2 did not decrease (*Figure 5—figure supplement 1A*), and qPCR confirmed that the cyAbrB2 binding signal increased in all positions tested (*Figure 5C*). ChIP-seq data and ChIP-qPCR data indicate that the boundary between cyAbrB2 binding region and cyAbrB2-free region became obscured when the cells entered microoxic conditions due to increased binding of cyAbrB2 to both cyAbrB2 binding and cyAbrB2-free region. The protein amount of cyAbrB2 was not altered on entry to the microoxic condition (*Figure 5—figure supplement 1B*). The cyAbrB2 binding signal around the transiently upregulated genes became less specific upon entry into microoxic conditions, consistent with the general tendency (*Figure 5B*). The amount of DNA immunoprecipitated by cyAbrB1 was also increased in the microoxic condition, and the protein amount was not increased (*Figure 5—figure supplement 2*).

## Sigma factors SigE and SigA are excluded from cyAbrB2 binding regions regardless of GC contents

We searched for SigE and SigA binding sites under aerobic and microoxic conditions (*Figure 6—figure supplement 1*, left and right, respectively). The SigE and SigA peaks identified in this study predominantly covered the previously identified peaks (*Figure 6—figure supplement 2*), reproducing the previous study's conclusion (*Kariyazono and Osanai, 2022*), i.e., SigE and the primary sigma factor SigA share localization on the promoters of housekeeping genes, but SigE exclusively binds to the promoters of its dependent genes. SigE and SigA binding peaks were significantly excluded from the cyAbrB2 binding regions (*Figure 6A and B*). SigE preferred the cyAbrB2-free region in the aerobic condition more than SigA did (odds ratios of SigE and SigA being in the cyAbrB2-free region were 4.88 and 2.74, respectively). CyAbrB2 prefers AT-rich regions, but no correlation was found between the GC content and binding signals of SigE and SigA (*Figure 3—figure supplement 4*). Thus, SigA and SigE are excluded from cyAbrB2 binding regions regardless of GC contents.

## Dynamics of sigma factors upon exposure to the microoxic condition

When cells entered microoxic conditions, the binding signals of SigA and SigE were changed, although most of their peaks observed under aerobic conditions were present under microoxic conditions (*Figure 6—figure supplement 1*). The preference of SigE for the cyAbrB2-free region was alleviated in the microoxic condition (*Figure 6A*). Next, we focused on sigma factor dynamics in transiently upregulated genes. SigE, but not SigA, binds at the TSS of *pntAB* under aerobic and microoxic conditions (*Figure 6C*, top). SigE binding summits were not identified at the TSSs of the *hox* and *nifJ* operons under aerobic conditions. However, the SigE-specific binding summit appeared at the TSS of *nifJ* when cells entered microoxic conditions (*Figure 6C*, middle). A bimodal peak of SigE was observed at the TSS of the *hox* operon in a microoxic-specific manner (*Figure 6C*, bottom panel). The downstream side of the bimodal SigE peak coincides with the SigA peak and the TSS of TU1715.

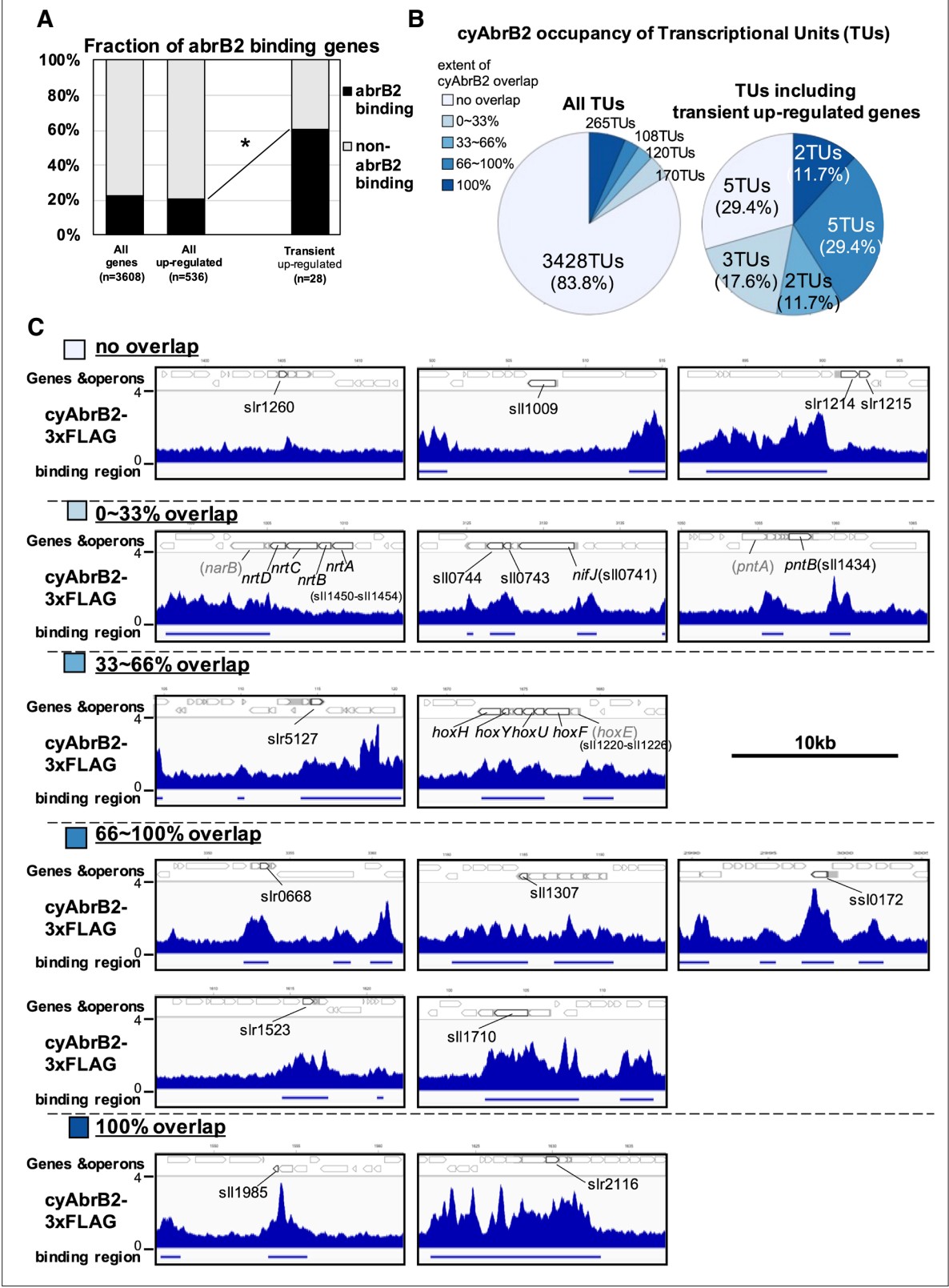

**Figure 4.** Transient up-regulated genes are enriched in cyAbrB2 binding regions. (**A**) Fraction of genes overlapped or non-overlapped with cyAbrB2 binding regions at the timepoints of aerobic conditions. Genes are classified according to *Figure 1—figure supplement 1*. Asterisk (*) denotes statistically significant enrichment compared with all upregulated genes tested by multiple comparisons of Fisher's exact test. (**B**) Pie charts of transcriptional units (TUs) classified by extent of overlapping with cyAbrB2 binding region. The left pie represents all TUs, and the right pie represents

*Figure 4 continued on next page*

*Figure 4 continued*

only TUs containing the transient upregulated genes. (**C**) Distribution of cyAbrB2 in the aerobic condition around transiently upregulated genes. Arrows with bold lines indicate transiently upregulated genes. Shaded arrows indicate operons whose data were obtained from a previous study. The bars below the graph indicate the binding regions of each protein. The black bar at the top of the figure indicates a length of 10 kbp.

Another side of the bimodal peak lacked SigA binding and was located at the TSS of the *hox* operon (marked with an arrow in *Figure 6C*), although the peak caller failed to recognize it as a peak. SigE binding without SigA on the promoters of *hox*, *nifj*, and *pntAB* is consistent with their SigE-dependent expression (*Figure 2B*).

### Chromatin conformation around *hox* operon and *nifJ* operon

We have shown that cyAbrB2 broadly binds to AT-rich genomic regions, including insertion element sequences, and represses expression (*Figure 3*). This is functionally similar to the NAPs (*Hołówka and Zakrzewska-Czerwińska, 2020*), which makes us hypothesize that cyAbrB2 modulates chromosomal conformation. Therefore, we conducted the chromatin conformation capture (3C) assay against wild-type and *cyabrb2Δ* strains at aerobic and microoxic conditions. qPCR was performed with unidirectional primer sets, where the genomic fragment containing *hox* operon and *nif* operon (hereinafter *hox* fragment and *nifJ* fragment, respectively) were used as bait (*Figure 7*).

First, focusing on the aerobic condition of wildtype (*Figure 7B*, solid line), the *hox* fragment interacted with its proximal downstream loci (loci (f) to (g)) and proximal upstream locus (locus (j)). The *hox* fragment also interacts with the distal downstream locus (locus (c)). Meanwhile, the *nifJ* fragment shows high interaction frequency with proximal upstream and downstream loci (*Figure 7G*, loci (i') and (j')), and a distal downstream locus (locus (g')) showed higher interaction frequency with *nifJ* fragment than proximal locus (h') did. The upstream regions of *nifJ* (loci (l') to (n') and (p')) showed comparable frequency with locus (g').

### The chromatin conformation is changed in *cyabrb2Δ* in some loci

Then we compared the chromatin conformation of wildtype and *cyabrb2Δ*. Although overall shapes of graphs did not differ, some differences were observed in wildtype and *cyabrb2Δ* (*Figure 7B and G*); interaction of locus (c) with *hox* region were slightly lower in *cyabrb2Δ* and interaction of loci (f') and (g') with *nifJ* region were different in wildtype and *cyabrb2Δ*. Note that the interaction scores exhibit considerable variability and we could not detect statistical significance at those loci.

### Changes of chromatin conformation upon microoxic condition

When the cells entered the microoxic condition, proximal loci interacted more frequently (*Figure 7D*, loci (f)–(h) and *Figure 7I*, loci (j') and (k')). This tendency was more apparent in *cyabrb2Δ* (*Figure 7E and J*). Furthermore, the interaction of *nifJ* upstream loci (l')–(n') increased in the microoxic condition in *cyabrb2Δ* but not wildtype (*Figure 7I and J*). The locus (c) and locus (j) interacted less frequently with *hox* fragment upon entry to the microoxic condition in the wildtype. While the interaction scores exhibit considerable variability, the individual data over time demonstrate declining trends of the wildtype at locus (c) and (j) (*Figure 7—figure supplement 1*). In Δ*cyabrb2*, by contrast, the interaction frequency of loci (c) and (j) was unchanged in the aerobic and microoxic conditions (*Figure 7E*). The interaction frequency of locus (c) in Δ*cyabrb2* was as low as that in the microoxic condition of wildtype, while that of locus (j) in Δ*cyabrb2* was as high as that in the aerobic condition of wildtype (*Figure 7B and C*). In summary, 3C analysis demonstrated cyAbrB2-dependent and independent dynamics of chromosomal conformation around the *hox and nifJ* operon in response to the microoxic condition (*Figure 8*).

## Discussion

### Physiological significance of transient upregulation of *hox* and *nifJ* operons

As the transcriptional change can alter the metabolic flow, the transcriptional upregulation of fermentative genes in response to the microoxic condition is expected to be adaptive for energy acquisition

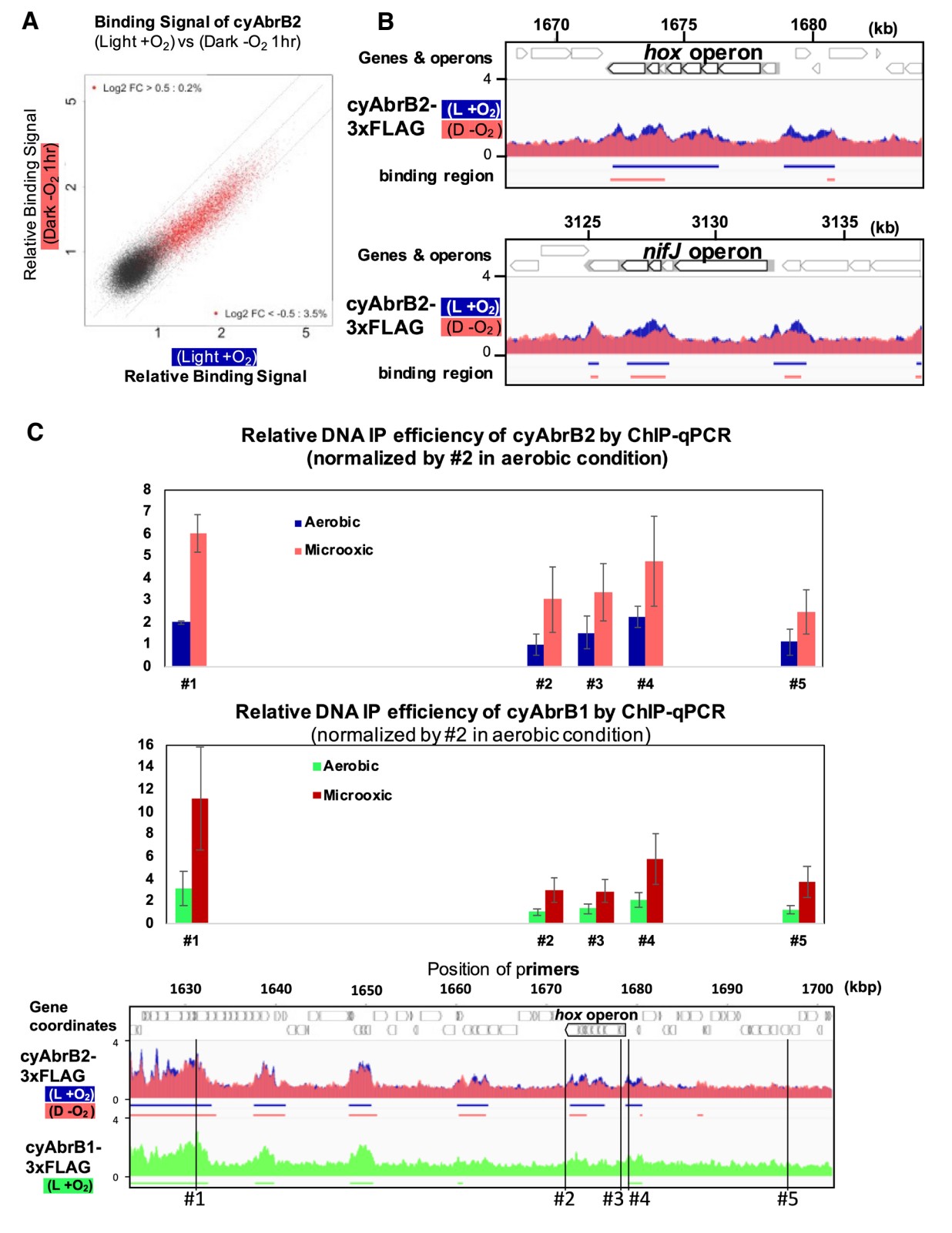

**Figure 5.** Changes of cyAbrB2 binding pattern on entry to the microoxic condition. (**A**) Scatter plot showing changes of the binding signal by 1 hr cultivation in the microoxic condition. The binding signal of each 100 bp window is plotted. Red dots are cyAbrB2 binding regions in either aerobic or microoxic conditions. The dotty lines indicate Log2 fold enrichment of 0.5, 0, and –0.5 between aerobic and microoxic conditions. (**B**) Distribution of cyAbrB2 around *hox* operon and *nifJ* operon. ChIP-seq data in aerobic (L + O$_2$) and dark microoxic (D − O$_2$) conditions are overlayed. The bars below

*Figure 5 continued*

the graph indicate the binding regions of each protein. (**C**) Quantification for IP efficiency of cyAbrB2 (top) and cyAbrB1 (middle) by qPCR in the aerobic and microoxic conditions. The position of primers and ChIP-seq data of cyAbrB2 are shown at the bottom. Scores are normalized by the IP% at position #2 in the aerobic condition. Error bars represent standard deviation (n=3).

The online version of this article includes the following figure supplement(s) for figure 5:

**Figure supplement 1.** Alteration of cyAbrB2 binding to genome under the microoxic condition.

**Figure supplement 2.** Alteration of cyAbrB1 binding to genome under the microoxic condition.

and the maintenance of redox balance. Our time-course transcriptome showed upregulation of several genes involved in catabolism upon exposure to the microoxic condition. The transient upregulation of *hox* and *nifJ* operons is distinctive among them (*Figure 1D*).

One reason for transient upregulation is probably the resource constraints of inorganic cofactors. Hydrogenase and PFOR (the product of *nifJ* gene) have iron-sulfur clusters, and hydrogenase requires nickel for its activity (*Uyeda and Rabinowitz, 1971*; *Vignais and Billoud, 2007*). Overexpression of the *hox* operon should be futile under physiological conditions without an adequate nickel supply (*Ortega-Ramos et al., 2014*).

Another significance for transient upregulation may be the reusability of fermentative products. Hydrogen, lactate, and dicarboxylic acids can be reused as the source of reducing power when cells return to aerobic conditions (*Appel et al., 2000*; *Katayama et al., 2022*; *Angermayr et al., 2016*). The substrate proton is abundant, but hydrogen is diffusive and difficult to store. Therefore, hydrogenase may favor fermentation initiation, and the reductive branch of TCA-producing dicarboxylic acids may become active subsequently. In fact, *citH/mdh* (sll0891) encoding a key enzyme of the reductive branch of TCA was classified as continuously upregulated genes in this study (*Figure 1C and D*).

## Mechanisms for transient expression mediated by SigE and cyAbrB2

SigE and cyAbrB2 can independently contribute to the transient transcriptional upregulation. This is evident as the single mutants, Δ*sigE* or Δ*cyabrb2*, maintained transient expression of *hoxF* and *nifJ* (*Figure 2—figure supplement 1*). We first discuss cyAbrB2 as the potential NAPs, and then the mechanism of transient upregulation mediated by cyAbrB2 and SigE will be discussed.

## cyAbrB2 is a novel nucleoid-associated protein of cyanobacteria

We have shown that cyAbrB2 broadly binds to AT-rich genomic regions, including IS elements (*Figure 3*). This is functionally similar to the histone-like nucleoid protein H-NS family, including H-NS in Enterobacteriaceae (*Navarre et al., 2007*; *Oshima et al., 2006*), and Lsr2 in *Mycobacteria* (*Gordon et al., 2010*). Like H-NS and Lsr2, cyAbrB2 may defend against exogenous DNA elements, which often have different GC content. Interestingly, Lsr2 controls genes responding to hypoxia, showing a functional analogy with cyAbrB2 (*Kołodziej et al., 2021*).

## The biochemistry of cyAbrB2 will shed light on the regulation of chromatin conformation in the future

H-NS proteins often cause bound DNA to bend, stiffen, and/or bridge (*Hołówka and Zakrzewska-Czerwińska, 2020*). DNA-bound cyAbrB2 is expected to oligomerize, based on the long tract of cyAbrB2 binding region in our ChIP-seq data. However, no biochemical data mentioned the DNA deforming function or oligomerization of cyAbrB2 in the previous studies, and preference for AT-rich DNA is not fully demonstrated in vitro (*Dutheil et al., 2012*; *Ishii and Hihara, 2008*; *Song et al., 2022*). Moreover, our 3C data did not support bridging at least in *hox* region and *nifJ* region, as the high interaction locus and cyAbrB2 binding region did not seem to correlate (*Figure 7*). Therefore, direct observation of the DNA-cyAbrB2 complex by atomic force microscopy is the solution in the future.

Not only DNA structural change but also the effect of the post-translational modification can be investigated by biochemistry. The previous studies report that cyAbrB2 is subject to phosphorylation and glutathionylation (*Spät et al., 2023*; *Sakr et al., 2013*), and pH and redox state alters cyAbrB1's affinity to DNA (*Song et al., 2022*). Those modifications might respond to environmental changes and be involved in transient expression. Overall, the biochemistry integrating assay conditions (PTM,

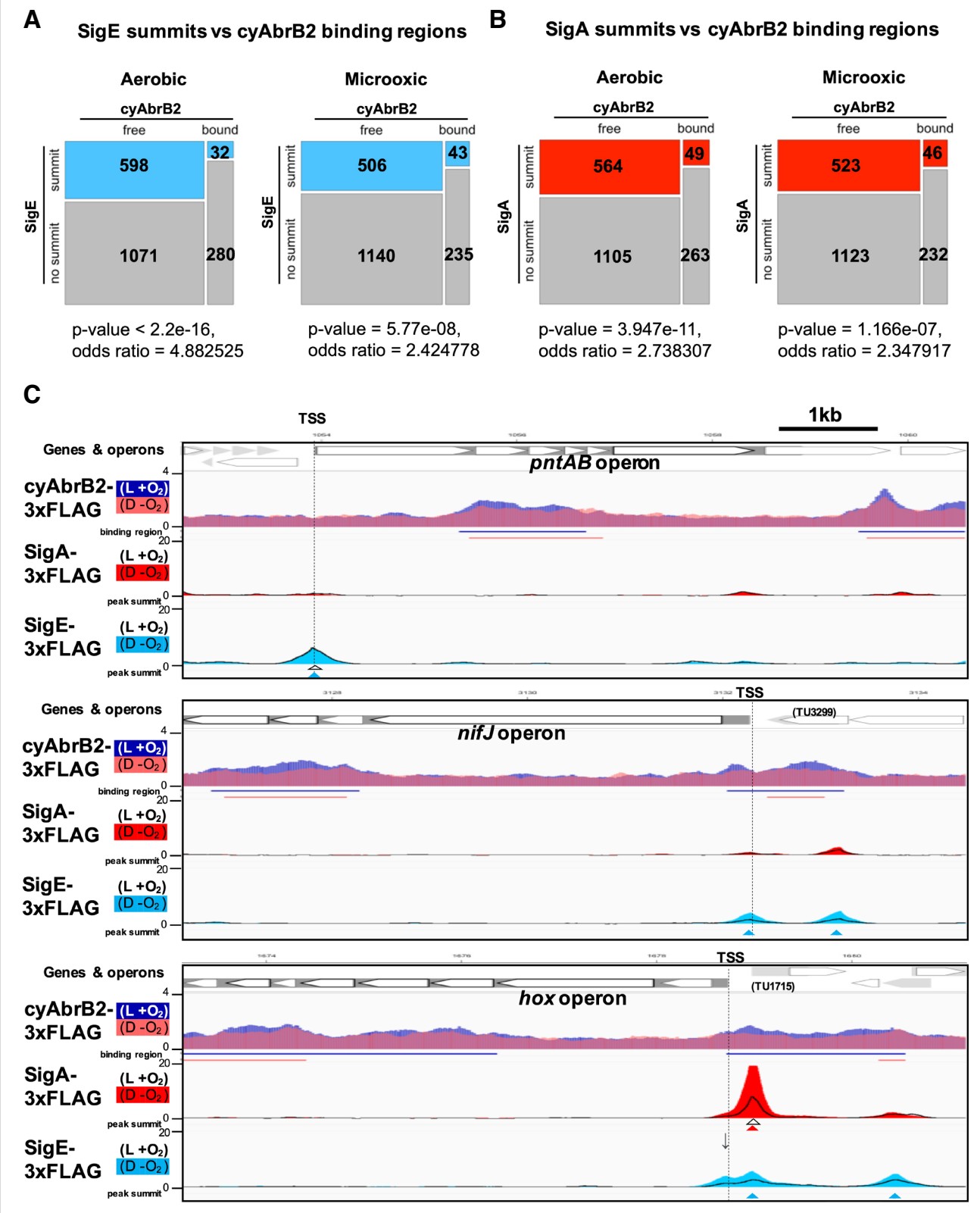

**Figure 6.** Sigma factors are excluded from cyAbrB2 binding regions. (**A and B**) Anti-co-occurrence of cyAbrB2 binding regions and sigma factors. Mosaic plots of cyAbrB2 binding regions and SigE peaks (**A**) or SigA binding peaks (**B**) are shown. Odds and p-values were calculated by Fisher's exact test. (**C**) Snapshots of ChIP-seq data for CyAabrB2, SigE, and SigA at the *nifJ* region (top) and *hox* region (bottom). ChIP-seq data for cyAbrB2, SigE, and SigA under aerobic and dark microoxic conditions are overlayed. ChIP-seq data of cyAbrB2 under aerobic and microoxic conditions are colored

*Figure 6 continued on next page*

*Figure 6 continued*

blue and pink, respectively. ChIP-seq data for SigE and SigA are shown in solid lines (aerobic conditions) and the area charts (microoxic conditions). The positions of transcription start sites (TSSs) were obtained from a previous study (*Kopf et al., 2014*) and indicated by vertical dotted lines. Open triangles indicate peak summits under aerobic conditions, and solid triangles indicate peak summits under microoxic conditions.

The online version of this article includes the following figure supplement(s) for figure 6:

**Figure supplement 1.** Changes of SigE and SigA distribution on the entry to the microoxic condition.

**Figure supplement 2.** Reproducibility of ChIP-seq data of SigA and SigE, compared with the previous study (*Kariyazono and Osanai, 2022*).

buffer condition, and cooperation with cyAbrB1) and output (DNA binding, oligomerization, and DNA structure) will deepen the understanding of cyAbrB2 as cyanobacterial NAPs.

## Cooperative and antagonistic function of cyAbrB1 and cyAbrB2

CyAbrB1, the homolog of cyAbrB2, may cooperatively work, as cyAbrB1 directly interacts with cyAbrB2 (*Yamauchi et al., 2011*), their distribution is similar, and they partially share their target genes for suppression (*Figure 3A* and *Figure 3—figure supplement 4*). The possibility of cooperation would be examined by the electrophoretic mobility shift assay of cyAbrB1 and cyAbrB2 as a complex. Despite their similar repressive function, cyAbrB1 and cyAbrB2 regulate *hox* expression in opposite directions, and their mechanism remains elusive. The stoichiometry of cyAbrB1 and cyAbrB2 bound to DNA fluctuates in response to the environmental changes (*Lieman-Hurwitz et al., 2009*), but there was no difference in the behavior of cyAbrB1 and cyAbrB2 around the *hox* region on entry to the microoxic condition.

## Localization pattern and function of cyAbrB2

Herein, we classified three types of binding patterns for cyAbrB2. The first is that cyAbrB2 binds a long DNA tract covering the entire gene or operon, represented by the insertion sequence elements. CyAbrB2 suppresses expression in this pattern (*Figure 3E*). In the second pattern, cyAbrB2 binds on promoter regions, such as *hox* operon and *nifJ*. The binding on those promoters fluctuates in response to environmental changes, thus regulating expression. This pattern also applies to the promoter of *sbtA* ($Na^+/HCO_3^-$ symporter), where cyAbrB2 is bound in a $CO_2$ concentration-dependent manner (*Lieman-Hurwitz et al., 2009*). The last one is cyAbrB2 binding in the middle or downstream of operons. The middle of *hox*, *pntAB*, and *nifJ* operons and the downstream of *nrt* operon are the cases (*Figure 4C*). Our data show that genes in the same operon separated by the cyAbrB2 binding region behave differently. In particular, *pntB* is classified as the transiently upregulated gene, while *pntA* is not, despite being in the same operon. This might be explained by the recent study which reported that cyAbrB2 affects the stability of mRNA transcribed from its binding gene (*Song et al., 2022*). The cyAbrB2-mediated stability of mRNA may also account for the decrease in transcript from transient upregulated genes at 4 hr of cultivation. Hereafter, we will focus on the mechanism of the second pattern, regulation by cyAbrB2 on the promoter.

## Insight into the regulation of *hox* and *nifJ* operon by cyAbrB2

Genome-wide analysis indicates that the cyAbrB2-bound region blocks SigE and SigA (*Figure 6A and B*). This is presumably because sigma factors recognize the promoter as a large complex of RNA polymerase. CyAbrB2 binds to the *hox and nifJ* promoter region and may inhibit access to RNA polymerase complex under aerobic conditions. When cells entered microoxic conditions, the boundaries of the cyAbrB2 binding region and cyAbrB2-free region became obscure (*Figure 5*), and SigE binding peaks on those promoters became prominent (*Figure 6C*). Notably, cyAbrB2 ChIP efficiency at the *hox* promoter is higher in the microoxic condition than in the aerobic condition (*Figure 5*). Hence, while the exclusion by cyAbrB2 occupancy on promoter inhibits containing RNA polymerase in the aerobic condition, it is also plausible that chromosomal conformation change governed by cyAbrB2 provides SigE-containing RNA polymerase with access to the promoter region (*Figure 8*). Our 3C result demonstrated that cyAbrB2 influences the chromosomal conformation of *hox* and *nifJ* region to some extent (*Figure 7*).

A recent study demonstrated that manipulating the expression of topoisomerase, which influences chromosomal conformational change through supercoiling, affects transcriptional properties in

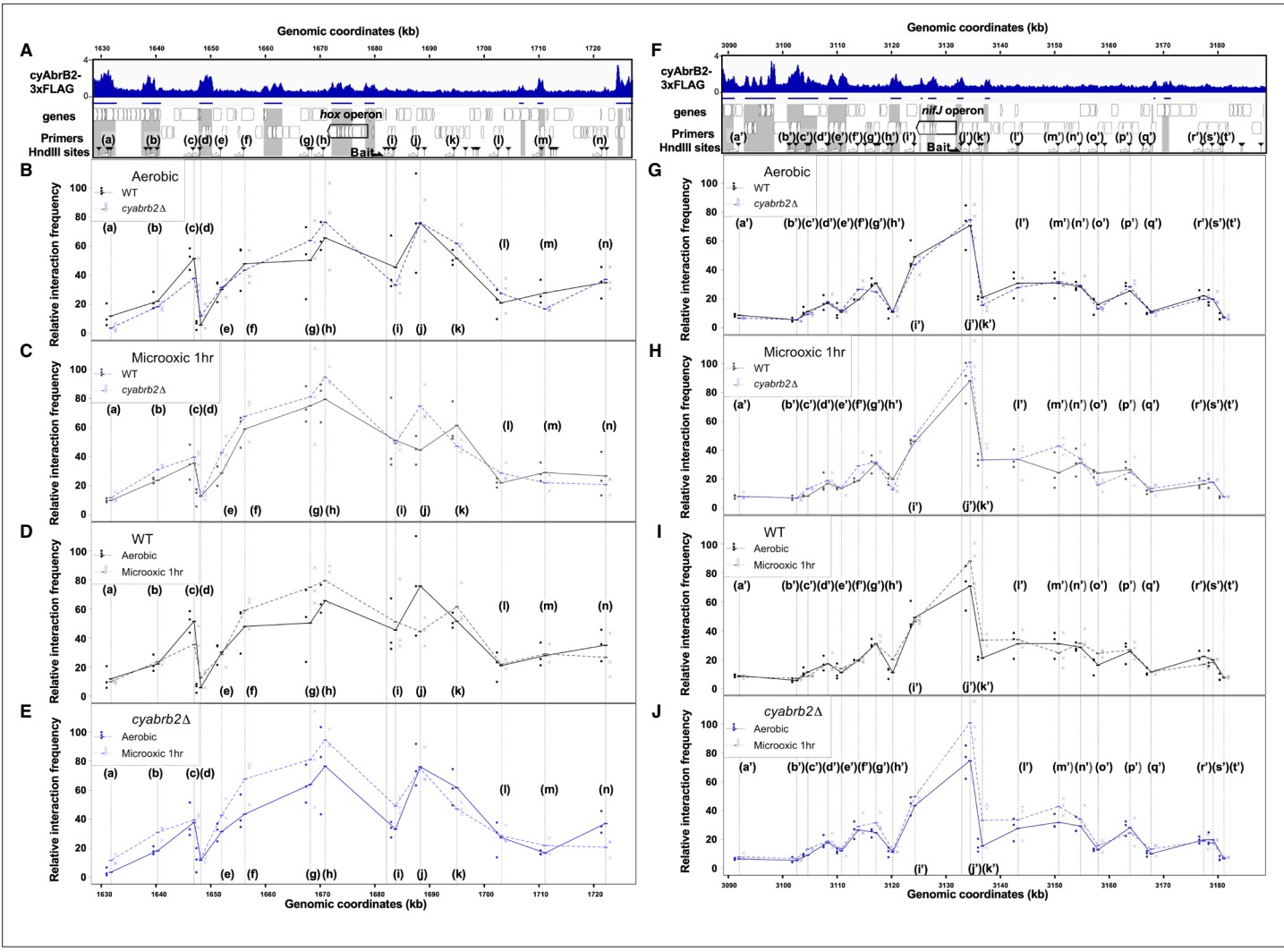

**Figure 7.** 3C analysis showed changes of DNA conformation around *hox* and *nifJ* operon on entry to microoxic condition and the impact of cyabrb2 deletion on DNA conformation. (**A and F**) Schematic diagram of 3C analysis around *hox* operon (**A**) and *nifJ* operon (**F**). In the panels (**A**) and (**F**), the black horizontal arrow shows the location of the bait primer, and white horizontal arrows ((a) to (n) in *hox* operon (**A**) and (a') to (t') in *nifJ* operon (**F**)) indicate loci where the interaction frequency with bait were assayed. Vertical black arrowheads indicate the position of HindIII sites. ChIP-seq data of cyAbrB2 in the aerobic condition is displayed in the bottom, and cyAbrB2 binding regions are marked with shade. (**B–E**) The line plot showing the interaction frequency of each locus with *hox* fragment. Two of data sets are presented; (**B**) wildtype vs Δ*cyabrb2* in aerobic condition, (**C**) wildtype vs Δ*cyabrb2* in 1 hr of microoxic condition, (**E**) wildtype in aerobic vs 1 hr of microoxic condition, and (**E**) Δ*cyabrb2* in aerobic vs 1 hr of microoxic condition are compared. (**G–J**) The line plot showing the interaction frequency of each locus with *nifJ* fragment. Two data sets are selected and presented; (**G**) wildtype vs Δ*cyabrb2* in aerobic condition, (**H**) wildtype vs Δ*cyabrb2* in 1 hr of microoxic condition, (**I**) wildtype in aerobic vs 1 hr of microoxic condition, and (**J**) Δ*cyabrb2* in aerobic vs 1 hr of microoxic conditions are compared. The line plots indicate the average interaction frequency over the random ligation (n=3). Individual data are plotted as dots.

The online version of this article includes the following figure supplement(s) for figure 7:

**Figure supplement 1.** Dynamics of individual 3C scores.

**Figure supplement 2.** The validation of unidirectional primer sets for 3C assay is shown in *Figure 7*.

cyanobacteria (*Behle et al., 2022*). Moreover, *Song et al., 2022* pointed out that DNA looping may inhibit transcription in cyanobacteria because artificial DNA looping by the LacI repressor of *Escherichia coli* can repress cyanobacterial transcription (*Camsund et al., 2014*). Thus, we infer conformation change detected by the present 3C experiment regulates expression of *hox* operon.

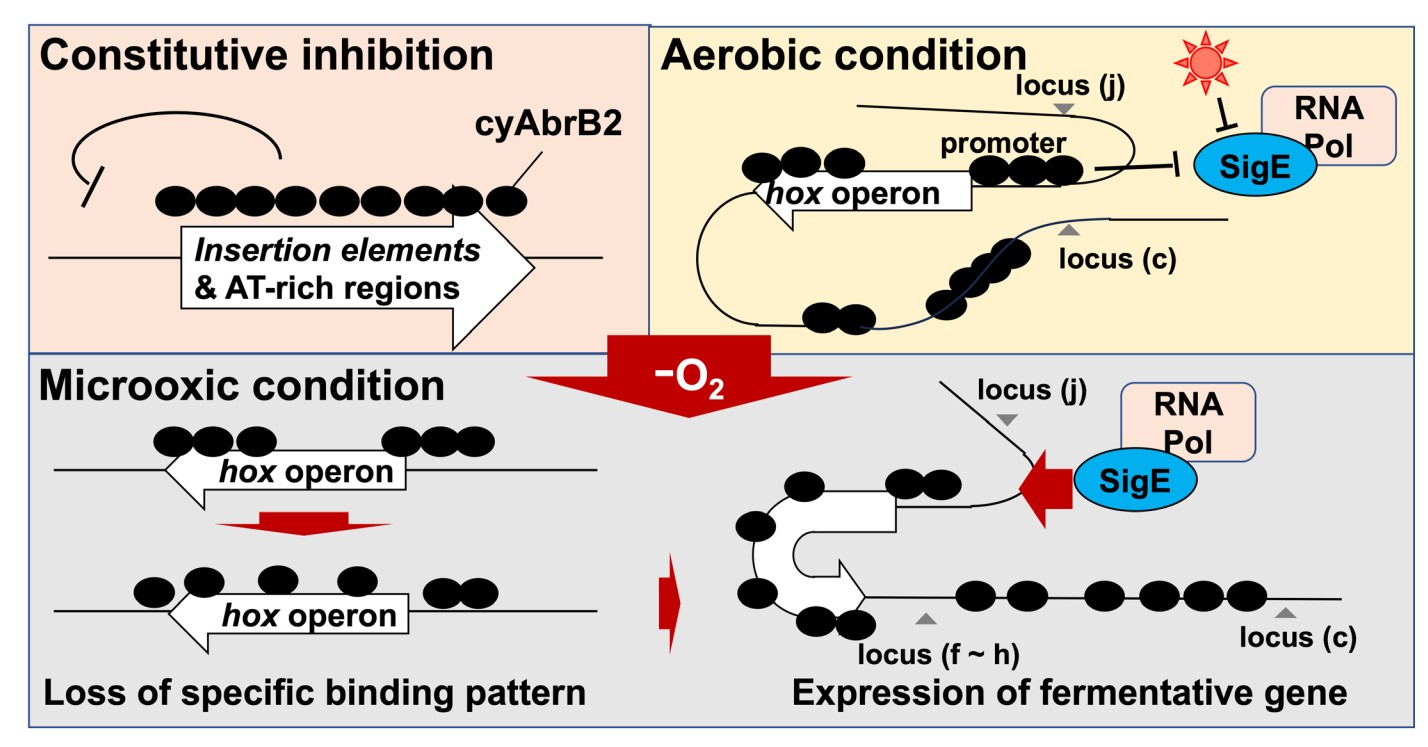

**Figure 8.** Schematic diagram of the dynamics of transcription factors governing fermentative gene expression.

### Generality for chromosomal conformation in cyanobacteria

Our 3C analysis revealed that local chromosomal conformation changes upon entry to the microoxic conditions (*Figure 8*). As cyAbrB2 occupies about 15% of the entire genome and globally regulates gene expression, cyAbrB2 likely governs the whole chromosomal conformation. Furthermore, the conformational changes by deletion of cyAbrB2 were limited, suggesting there are potential NAPs in cyanobacteria yet to be characterized. It is speculated that conformational change of the entire chromosome occurs to deal with many environmental stresses.

### The sigE-mediated mechanism for the transient expression

One possible SigE-mediated mechanism for transient expression is the post-transcriptional activation and degradation of SigE in the dark, i.e., SigE is sequestered by anti-sigma factor under light conditions and released under dark (*Osanai et al., 2009*), enabling acute transcription of *hox* operon and *nifJ*. Transcripts of *sigE* were continuously downregulated in our time-course transcriptome, while *sigB* (sll0306) and *sigC* (sll0184) were classified as continuous upregulated genes (*Table 2*). It is possible

**Table 2.** Fold changes of transcripts from *sigA, sigB, sigC, sigD,* and *sigE*.

| Sigma factor | Locus | 0 hr vs 1 **hr** | | 1 hr vs 4 **hr** | |
| --- | --- | --- | --- | --- | --- |
| | | Log2FC | FDR | Log2FC | FDR |
| SigA | slr0653 | −0.873248 | 0.00766486 | −0.0013514 | 0.99797563 |
| SigB | sll0306 | 1.38098826 | 8.42E-06 | 0.77453605 | 0.04057775 |
| SigC | sll0184 | 2.97101055 | 1.75E-16 | 1.30743549 | 0.00067892 |
| SigD | sll2012 | 0.4701823 | 0.1498473 | −0.4522181 | 0.32402556 |
| SigE | sll1689 | −1.9111759 | 1.96E-11 | −1.1223298 | 0.00633142 |

Data is extracted from **Supplementary file 1d**.

that upregulated SigB and SigC outcompete SigE in prolonged incubation under microoxic conditions. Finally, SigE is degraded under dark within 24 hr (*Iijima et al., 2015*).

Another reason for the microoxic specific expression may exist in the sequence of the *hox* promoter. We previously determined the consensus sequence of –10 element for SigE regulon in the aerobic condition as 'TANNNT', where N is rich in cytosine (*Kariyazono and Osanai, 2022*). The –10 sequence of the *hox* promoter 'TAACAA' (*Oliveira and Lindblad, 2005*) deviates from the consensus, and no hexamer precisely fitting the consensus is found in the *nifJ* promoter. This deviation can inhibit SigE from binding during aerobic conditions, aside from cyAbrB2-mediated inhibition. Under the microoxic condition, transcription factors LexA (*Oliveira and Lindblad, 2005*) and Rre34 (*Summerfield et al., 2011*) may aid SigE binding to the promoter of *hox* and *nifJ*, respectively.

Moreover, SigE seems susceptible to the blocking from cyAbrB2 during the aerobic condition compared with SigA. This is supported by the odds ratio of SigE being in the cyAbrB2-free region was higher than that of SigA in the aerobic condition (*Figure 6A and B*). The higher exclusion pressure of cyAbrB2 on SigE may contribute to sharpening the transcriptional response of hox and nifJ on entry to microoxic conditions. Overall, multiple environmental signals are integrated into the *hox* and *nifJ* promoter through the cyAbrB2 and SigE dynamics.

# Materials and methods

## Key resources table

| Reagent type (species) or resource | Designation | Source or reference | Identifiers | Additional information |
|---|---|---|---|---|
| Strain, strain background (*Synechocystis* sp. PCC6803) | Wildtype | https://doi.org/10.1016/0076-6879(88)67088-1 | GT | |
| Strain, strain background (*Synechocystis* sp. PCC6803) | Δ*sigE*::KmR | https://doi.org/10.1074/jbc.M505043200 | G50 | |
| Strain, strain background (*Synechocystis* sp. PCC6803) | SigA-8His-KmR | https://doi.org/10.1111/tpj.15687 | KR93 | |
| Strain, strain background (*Synechocystis* sp. PCC6803) | SigA-3FLAG-KmR | https://doi.org/10.1111/tpj.15687 | KR94 | |
| Strain, strain background (*Synechocystis* sp. PCC6803) | Δ*cyabrb2*::KmR | In this study | KR340 | The genome of GT strain was manipulated by the transformation of the plasmid VK203 |
| Strain, strain background (*Synechocystis* sp. PCC6803) | cyAbrB(sll0359)–3xFLAG-KmR | In this study | KR338 | The genome of GT strain was manipulated by the transformation of the plasmid VK200 |
| Strain, strain background (*Synechocystis* sp. PCC6803) | cyAbrB2(sll0822)–3xFLAG-KmR | In this study | KR339 | The genome of GT strain was manipulated by the transformation of the plasmid VK201 |
| Strain, strain background (*Synechocystis* sp. PCC6803) | Δ*cyabrB2*::KmR Δ*sigE*::CmR | In this study | KR359 | The genome of G50 strain was manipulated by the transformation of the plasmid VK82 |
| Recombinant DNA reagent | *sigE*ΔCmR | In this study | VK82 | Plasmid backbone:pTA2 (Toyobo), available upon request |
| Recombinant DNA reagent | AbrB1-3F-KmR | In this study | VK200 | Plasmid backbone:pTA2 (Toyobo), available upon request |
| Recombinant DNA reagent | AbrB2-3F-KmR | In this study | VK201 | Plasmid backbone:pTA2 (Toyobo), available upon request |
| Recombinant DNA reagent | *cyabrB2*ΔKmR | In this study | VK203 | Plasmid backbone:pTA2 (Toyobo), available upon request |
| Antibody | Anti-FLAG | Sigma-aldrich | F1804 | RRID:AB_262044 For immunoprecipitation |
| Antibody | Anti-FLAG (alkaline phosphatase conjugated) | Sigma-aldrich | A9469 | RRID:AB_439699 For western blot (1:20,000) |

## Bacterial strains and plasmids

The glucose-tolerant strain of *Synechocystis* sp. PCC 6803 (*Williams, 1988*) was used as a wild-type strain in this study. The *sigE* (sll1689)-disrupted strain (G50), SigE FLAG-tagged strain, and

SigA FLAG-tagged strain were constructed in a previous study (*Osanai et al., 2005*; *Kariyazono and Osanai, 2022*). Disruption and epitope tagging of *cyabrb1*(sll0359) and *cyabrb2*(sll0822) were performed by homologous double recombination between the genome and PCR fragment (*Williams, 1988*). The resulting transformants were selected using three passages on BG-11 plates containing 5 µg/mL kanamycin. Genomic PCR was used to confirm the insertion of epitope tag fragments and gene disruption (*Figure 3—figure supplement 3*). Key resources table and *Supplementary file 1* contain the cyanobacterial strains, oligonucleotides, and plasmids used in this study.

## Aerobic and microoxic culture conditions

For aerobic conditions, cells were harvested after 24 hr cultivation in HEPES-buffered BG-11$_0$ medium (*Stanier et al., 1979*), which was buffered with 20 mM HEPES-KOH (pH 7.8) containing 5 mM NH$_4$Cl under continuous exposure to white light (40 µmol/m$^2$/s) and bubbled with air containing 1% CO$_2$ (final OD$_{730}$=1.4–1.8). For the dark microoxic culture, the aerobic culture cell was concentrated to an OD$_{730}$ of 20 with the centrifuge and resuspended in the culture medium. The concentrated cultures were poured into vials, bubbled with N$_2$ gas, and sealed. The sealed vials were shaded and shaken at 30°C for the described times.

## Antibodies and immunoblotting

Sample preparation for immunoblotting was performed as previously described (*Kariyazono and Osanai, 2022*), and FLAG-tagged proteins were detected by alkaline-phosphatase-conjugated anti-FLAG IgG (A9469, Sigma-Aldrich, St. Louis, MO, USA) and 1-Step NBT/BCIP substrate solution (Thermo Fisher Scientific, Waltham, MA, USA).

## RNA isolation

Total RNA was isolated with ISOGEN (Nippon Gene, Tokyo, Japan) following the manufacturer's instructions and stored at −80°C until use. The extracted RNA was treated with TURBO DNase (Thermo Fisher Scientific) for 1 hr at 37°C to remove any genomic DNA contamination. We confirmed that the A260/A280 of the extracted RNA was >1.9 by NanoDrop Lite (Thermo Fisher Scientific). We prepared triplicates for each timepoint for the RNA-seq library. RT-qPCR was performed as described elsewhere (*Iijima et al., 2015*).

## ChIP assay

Two biological replicates were used for each ChIP-seq experiment, and one untagged control ChIP was performed. ChIP and qPCR analyses were performed using the modified version of a previous method (*Kariyazono and Osanai, 2022*). FLAG-tagged proteins were immunoprecipitated with FLAG-M2 antibody (F1804 Sigma-Aldrich) conjugated to protein G dynabeads (Thermo Fisher Scientific).

## Library preparation and next-generation sequencing

For the ChIP-seq library, input and immunoprecipitated DNA were prepared into multiplexed libraries using NEBNext Ultra II DNA Library Prep Kit for Illumina (New England Biolabs, Ipswich, MA, USA). For the RNA-seq library, isolated RNA samples were deprived of ribosomal RNA with Illumina Ribo-Zero Plus rRNA Depletion Kit (Illumina, San Diego, CA, USA) and processed into a cDNA library for Illumina with the NEBNext Ultra II Directional RNA Library Prep Kit for Illumina (New England Biolabs). Dual-index primers were conjugated with NEBNext Multiplex Oligos for Illumina (Set1, New England Biolabs). We pooled all libraries, and the multiplexed libraries were dispatched to Macrogen Japan Inc and subjected to paired-end sequencing with HiSeqX. Adapter trimming and quality filtering of raw sequence reads were conducted with fastp (ver. 0.21.0) (*Chen et al., 2018*) under default conditions. The paired-end sequences were mapped onto the *Synechocystis* genome (ASM972v1) using Bowtie2 (*Langmead and Salzberg, 2012*) (ver. 2.4.5 paired-end). *Supplementary file 1* contains the read counts that passed via fastp quality control and were mapped by Bowtie2.

## RNA-seq analysis

Mapped reads were counted by HT-seq count (ver. 2.0.2) (*Anders et al., 2015*) for the GFF file of ASM972v1, with the reverse-strandedness option. EdgeR package (ver. 3.40.1) (*Robinson et al., 2010*) was used to perform the differential expression analysis. Fold changes in expression and FDR

were used for gene classification. *Supplementary file 1* contains fold change in gene expression calculated by edgeR.

## Genome-wide analyses

Peaks were called using the MACS3 program (ver. 3.0.0b1) (*Zhang et al., 2008*). For paired-end reads for SigE, SigA, and untagged control ChIP, narrow peaks were called with <1e−20 of the q-value cut-off and '--call-summits' options. The peak summits from two replicates and the untagged control were merged if summits were located within 40 bp of each other. Peak summits identified in both replicates but not in the control were considered for further analysis. The midpoint of the peak summits for the two merged replicates was further analyzed.

Broad peak calling methods were applied to paired-end reads for cyAbrB2, cyAbrB1, and untagged control ChIP using the '–broad' option, with a q-value cut-off of <0.05 and a q-value broad cut-off of <0.05. The intersection of broad peaks from two replicates, excluding those called by the control, was used in subsequent analyses.

The positions of the TSS, including internal start sites, were obtained as reported by *Kopf et al., 2014*. The read count, merging, and intersection of the binding region were calculated using BEDTools (ver. 2.30.0) (*Quinlan and Hall, 2010*).*Supplementary file 1* contain SigA and SigE peaks and the broad binding regions of cyAbrB2 and cyAbrB1, respectively.

Binding signals in every 100 bp bin for scatter plots were calculated as (IP read counts within 100 bp window)/(input read counts within 100 bp window) * (total input read counts/total IP read counts). GC contents were calculated within 500 bp in 100 bp sliding windows by seqkit (ver. 2.3.0) (*Shen et al., 2016*).

## Genome extraction, digestion, and ligation for 3C assay

A 3C assay was conducted based on the previous prokaryotic Hi-C experiment (*Takemata et al., 2019*; *Takemata and Bell, 2021*), with certain steps modified. To begin, *Synechocystis* were fixed with 2.5% formaldehyde for 15 min at room temperature. Fixation was terminated by adding a final concentration of 0.5 M of glycine, and cells were stored at –80°C until use. Fixed cells were disrupted using glass beads and shake master NEO (Bio Medical Science, Tokyo, Japan), following the previous study's instructions for preparing cell lysate for ChIP. The lysates were incubated with buffer containing 1 mM Tris-HCl (pH 7.5), 0.1 mM EDTA, and 0.5% SDS for 10 min at room temperature, and 1% Triton X-100 quenched SDS. Genomes in the cell lysate were digested by 600 U/mL of HindIII (Takara Bio, Shiga, Japan) for 4 hr at 37°C, and RNA in the lysate was simultaneously removed by 50 μg/mL of RNaseA (Nippon Genetics, Tokyo, Japan). The digestion was terminated by adding 1% SDS and 22 mM EDTA. The fill-in reaction and biotin labeling steps were omitted from the procedure. The digested genomes were diluted by ligation buffer containing 1% Triton X-100 to the final concentration of approximately 1 μg/mL and incubated for 10 min at room temperature. Ligation was performed with 2 U/mL of T4 DNA ligase (Nippon Gene) overnight at 16°C. Crosslinking was reversed under 65°C for 4 hr in the presence of 2.5 mg/mL of proteinase K (Kanto Chemical, Tokyo, Japan), and DNA was purified with the phenol-chloroform method and ethanol precipitation method.

## Preparation of calibration samples for 3C qPCR

Based on a previous study, calibration samples for possible ligated pairs were prepared in parallel with 3C ligation (*Abou El Hassan and Bremner, 2009*). In brief, the purified genome of *Synechocystis* was digested by HindIII, and DNA was purified with the phenol-chloroform and ethanol precipitation. Purified DNA was dissolved into the ligation buffer at a concentration of about 600 ng/μL and ligated with 2 U/mL of T4 DNA ligase at 16°C overnight.

## Quantification of crosslinking frequency for 3C assay

Before the real-time PCR assay, we confirmed that each primer set amplified single bands in a ligation-dependent manner by GoTaq Hot Start Green Master Mix (Promega, Madison, WI, USA) (*Figure 7— figure supplement 2*). Real-time PCR was performed with StepOnePlus (Applied Biosystems, Foster City, CA, USA) and Fast SYBR Green Master Mix (Thermo Fisher Scientific) according to the manufacturer's instructions. Interaction frequency was calculated by ΔΔCt method using dilution series of calibration samples described above. We confirmed each primer set amplified DNA fragment with a

unique Tm value. The amount of the bait fragment containing *hox* operon were quantified and used as an internal control. *Supplementary file 1a* contains the list of primers used in the 3C quantification. Interaction frequency for each primer position was calculated as the relative abundance of ligated fragments against the calibration samples and normalized among samples by internal control.

### Statistical analysis

Statistical analyses were performed with R version 4.2.2 (*R Development Core Team, 2021*). The 'fisher.test' function was used for Fisher's exact test, and p-values< 0.05 were denoted as asterisks in the figure. Multiple comparisons of Fisher's exact test were conducted using 'fisher.Multcomp' function in the RVAideMemoire package (*Hervé, 2022*), where p-values were adjusted by the 'fdr' method and FDRs<0.05 are shown in the figures. Multiple comparisons of the Wilcoxon-rank test were conducted by 'pairwise.wilcox.test', and p-values were adjusted by the 'fdr' method. Adjusted p-values<0.05 are shown in the figure. The correlation coefficient was calculated with the 'cor' function. GSEA was performed by culsterPlofiler package *Wu et al., 2021* in R with p-value cut-off of 0.05. The enriched pathways detected by GSEA are listed in *Supplementary file 1*.

## Acknowledgements

This study was supported by the following grants to TO: Grant-in-Aid for Scientific Research (B) (grant no. 20H02905), JST-ALCA of the Japan Science and Technology Agency (grant number JPMJAL1306), the Asahi Glass Foundation, and JST-GteX (grant number JPMJGX23B0). We thank Dr. Kohki Yoshimoto for providing laboratory instruments and Ms. Kaori Iwazumi for the support of bacterial culture and the medium preparation.

## Additional information

### Funding

| Funder | Grant reference number | Author |
| --- | --- | --- |
| Japan Society for the Promotion of Science | 20H02905 | Takashi Osanai |
| Japan Science and Technology Agency | 10.52926/jpmjal1306 | Takashi Osanai |
| Japan Science and Technology Agency GteX | JPMJGX23B0 | Takashi Osanai |
| Asahi Glass Foundation | | Takashi Osanai |

The funders had no role in study design, data collection and interpretation, or the decision to submit the work for publication.

### Author contributions

Ryo Kariyazono, Conceptualization, Investigation, Visualization, Writing - original draft, Writing - review and editing; Takashi Osanai, Supervision, Funding acquisition, Project administration, Writing - review and editing

### Author ORCIDs

Ryo Kariyazono https://orcid.org/0000-0002-7591-3117
Takashi Osanai https://orcid.org/0000-0003-2229-6028

Reviewer #3 (Public Review): https://doi.org/10.7554/eLife.94245.3.sa1
Author response https://doi.org/10.7554/eLife.94245.3.sa2

## Additional files

### Supplementary files

• Supplementary file 1. Oligonucleotides used in this study and the summary of NGS analysis. (a) Oligonucleotides used in this study. (b) Numbers and percentages of NGS reads passed the processes. (c–e) Log2FC, LogCPM, LR, p-value, and false discovery rate (FDR) calculated by edgeR lrt method. (c) Processed data from time-course transcriptome for GT strain. (d) Processed data from the comparison between GT and sigEΔ strain in each timepoints. (e) Processed data from the comparison between GT and cyabrb2Δ strain in each timepoints. (f) List of SigE binding summit in the aerobic condition from ChIP-seq data. (g) List of SigE binding summit in the microoxic condition from ChIP-seq data. (h) List of SigA binding summit in the aerobic condition from ChIP-seq data. (i) List of SigA binding summit in the microoxic condition from ChIP-seq data. (j) List of cyAbrB2 binding region in the aerobic condition from ChIP-seq data. (k) List of cyAbrB2 binding region in the microoxic condition from ChIP-seq data. (l) List of cyAbrB1 binding region in the aerobic condition from ChIP-seq data. (m) Raw result of gene set enrichment analysis of time-course transcriptome (vs the aerobic condition).

• MDAR checklist

• Source data 1. Uncropped images for *Figure 3—figure supplement 2 and 3*, *Figure 5—figure supplement 1 and 2*, and *Figure 7—figure supplement 2*.

### Data availability

ChIP sequencing and RNA sequencing reads were deposited in the Sequence Read Archive (accession ID: PRJNA956842).

The following dataset was generated:

| Author(s) | Year | Dataset title | Dataset URL | Database and Identifier |
|---|---|---|---|---|
| Kariyazono R, Osanai T | 2023 | Time course transcriptome of Synechocystis sp. PCC6803 under the microoxic conditions | https://www.ncbi.nlm.nih.gov/bioproject/?term=PRJNA956842 | NCBI BioProject, PRJNA956842 |

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
