## [Editor Report · eLife assessment]

The authors provide **solid** data on a functional investigation of potential nucleoid-associated proteins and the modulation of chromosomal conformation in a model cyanobacterium. These **valuable** findings will be of interest to the chromosome and microbiology fields. Additional analysis and the tempering of conclusions has helped to improve the work, although further refinement remains possible.

---

## [Referee Report · Reviewer #3 (Public Review)]

This work probes the control of the hox operon in the cyanobacterium Synechocystis, where this operon directs the synthesis of a bidirectional hydrogenase that functions to produce hydrogen. In assessing the control of the hox system, the authors focused on the relative contributions of cyAbrB2, alongside SigE (and to a lesser extent, SigA and cyAbrB1) under both aerobic and microoxic conditions. In mapping the binding sites of these different proteins, they discovered that cyAbrB2 bound many sites throughout the chromosome, repressed many of its target genes, and preferentially bound regions that were (relatively) rich in AT-residues. These characteristics led the authors to consider that cyAbrB2 may function as a nucleoid-associated protein (NAP) in Synechocystis, given the functional similarities with other NAPs like H-NS. They assessed the local chromosome conformation in both wild type and cyabrB2 mutant strains at multiple sites within a 40 kb window on either side of the hox locus, using a region within the hox operon as bait. They concluded that cyAbrB2 functions as a nucleoid associated protein that influences the activity of SigE through its modulation of chromosome architecture.

The authors approached their experiments carefully, and the data were generally very clearly presented. At the same time, the overall work contains many lines of inquiry and different protein investigations that in some ways made it more challenging to identify the overall take-away message(s).

Based on the data presented, the authors make a strong case for cyAbrB2 as a nucleoid-associated protein, given the multiple ways in which is seems to function similarly to the well-studied *Escherichia coli* H-NS protein. They now provide additional commentary that relates cyAbrB2 with other nucleoid-associated proteins.

Previous work had revealed a role for SigE in the control of hox cluster expression, which nicely justified its inclusion (and focus) in this study. The focus on cyAbrB2 is also well-justified, given previous reports of its control of hox expression; however, it shares binding sites with an essential homologue cyAbrB1. Interestingly, while the B1 protein appears to bind similar sites, instead of repressing hox expression, it is known as an activator of this operon. If the information on cyAbrB1 is retained in the manuscript, it would be important to consider how cyAbrB1 activity might influence the results described here (although the authors could also consider removing the cyAbrB1 information to help improve the focus of the manuscript).

---

## [Author Response]

The following is the authors’ response to the original reviews.

**eLife assessment**
The authors provide solid data on a functional investigation of potential nucleoid-associated proteins and the modulation of chromosomal conformation in a model cyanobacterium. While the experiments presented are convincing, the manuscript could benefit from restructuring towards the precise findings; alternatively, additional data buttressing the claims made would significantly enhance the study. These valuable findings will be of interest to the chromosome and microbiology fields.

We appreciate editors for taking time for assessment and reviewers for giving critical suggestions. Both reviewers were concerned about our interpretation of 3C data, and Reviewer #2 suggested the biochemistry of cyAbrB2 to reinforce our claim. We agree with the concern and suggest editors add a sentence “How cyAbrB2 affects chromosome structure is still elusive from this study, and the biochemical assays are needed in the future experiment.” to the eLife assessment.

The major revision points are the following;

Reconstruction of Figures

Previous Figure 5E has been omitted

Additional 3C data on the nifJ region

Rephrasing the conclusion of 3C data

Additional discussion on cyAbrB2 and NAPs

**Reviewer #1 (Public Review):**
Strength:At first glance, I had a very positive impression of the overall manuscript. The experiments were well done, the data presentation looks very structured, and the text reads well in principle.Weakness:Having a closer look, the red line of the manuscript is somewhat blurry. Reading the abstract, the introduction, and parts of the discussion, it is not really clear what the authors exactly aim to target. Is it the regulation of fermentation in cyanobacteria because it is under-investigated? Is it to bring light to the transcriptional regulation of hydrogenase genes? The regulation by SigE? Or is it to get insight into the real function of cyAbrB2 in cyanobacteria? All of this would be good of course. But it appears that the authors try to integrate all these aspects, which in the end is a little bit counterintuitive and in some places even confusing. From my point of view, the major story is a functional investigation of the presumable transcriptional regulator cyAbrB2, which turned out to be a potential NAP. To demonstrate/prove this, the hox genes have been chosen as an example due to the fact that a regulatory role of cyAbrB2 has already been described. In my eyes, it would be good to restructure or streamline the introduction according to this major outcome.

As you pointed out, the major focus of this study is cyAbrB2 as a potential NAPs. To focus on NAPs, we simplified the first paragraph of the discussion (ll.246-263) and added the section comparing cyAbrB2 with other known NAPs (11.269-299). To emphasize the description of cyAbrB2, we also rearranged the figures and divided the analysis on cyAbrB2 ChIP into two figures. We reduced the first paragraph of the introduction but mostly preserved the composition of the introduction to keep the general to specific pattern, even though the manuscript is blurry.

Points to consider:The authors suggest that the microoxic condition is the reason for the downregulation of e.g. photosynthesis (l.112-114). But of course, they also switched off the light to achieve a microoxic environment, which presumably is the trigger signal for photosynthesis-related genes. I suggest avoiding making causal conclusions exclusively related to oxygen and recommend rephrasing (for example, "were downregulated under the conditions applied").

We agree with this point. We rephrased l.114 to “by the transition to dark microoxic conditions from light aerobic conditions” (ll.108-109).

The authors hypothesized that cyAbrB2 modulates chromosomal conformation and conducted a 3C analysis. But if I read the data in Figure 5B & C correctly, there is a lot of interaction in a range of 1650 and 1700 kb, not only at marked positions c and j. Positions c and j have been picked because it appears that cyAbrB2 deletion impacts this particular interaction. But is it really significant? In the case of position j the variation between the replicates seems quite high, in the case of position c the mean difference is not that high. Moreover, does all this correlate with cyAbrB2 binding, i.e. with positions of gray bars in panel A? If this was the case, the data obtained for the cyabrB2 mutant should look totally different but they are quite similar to WT. That's why the sentence "By contrast, the interaction frequency in Δcyabrb2 mutant were low and unchanged in the aerobic and microoxic conditions" does not fit to the data shown. But I have to mention that I am not an expert in these kinds of assays. Nevertheless, if there is a biological function that shall be revealed by an experiment, the data must be crystal clear on that. At least the descriptions of the 3C data and the corresponding conclusions need to be improved. For me, it is hard to follow the authors' thoughts in this context.

According to your suggestion, we again have carefully observed the 3C data. Furthermore, we conducted an additional 3C experiment on nifJ region (Figures 7F-J). Then we admit we had overinterpreted the 3C data. Therefore, we rewrote the result and discussion of the 3C assay in line with the data (ll.220-245) and removed the previous Figure 5E. Following are individual responses.

Positions c and j have been picked because it appears that cyAbrB2 deletion impacts this particular interaction. But is it really significant?

We could not find statistically significant differences at locus c and j. Therefore, we added this in the result section “Note that the interaction scores exhibit considerable variability and we could not detect statistical significance at those loci.” (ll.231-232)

does all this correlate with cyAbrB2 binding, i.e. with positions of gray bars in panel A?

As you are concerned, interaction frequency and cyAbrB2 binding do not correlate. Therefore, we withdraw the previous claim and stated as follows; “Moreover, our 3C data did not support bridging at least in hox region and nifJ region, as the high interaction locus and cyAbrB2 binding region did not seem to correlate (Figure 7).” (ll.280-282)

If this was the case, the data obtained for the cyabrB2 mutant should look totally different but they are quite similar to WT.

We rewrote it as follows; “Then we compared the chromatin conformation of wildtype and cyabrb2∆. Although overall shapes of graphs did not differ, some differences were observed in wildtype and cyabrb2∆ (Figures 7B and 7G); interaction of locus (c) with hox region were slightly lower in cyabrb2∆ and interaction of loci (f’) and (g’) with nifJ region were different in wildtype and cyabrb2∆. Note that the interaction scores exhibit considerable variability and we could not detect statistical significance at those loci.” (ll.228-232)

That's why the sentence "By contrast, the interaction frequency in Δcyabrb2 mutant were low and unchanged in the aerobic and microoxic conditions" does not fit to the data shown.

We rewrote the sentence as follow; “While the interaction scores exhibit considerable variability, the individual data over time demonstrate declining trends of the wildtype at locus (c) and (j) (Figure S8). In ∆cyabrb2, by contrast, the interaction frequency of loci (c) and (j) was unchanged in the aerobic and microoxic conditions (Figure 7E). The interaction frequency of locus (c) in ∆cyabrb2 was as low as that in the microoxic condition of wildtype, while that of locus (j) in ∆cyabrb2 was as high as that in the aerobic condition of wildtype (Figures 7B and 7C).” (ll.238-243)

The figures are nicely prepared, albeit quite complex and in some cases not really supportive of the understanding of the results description. Moreover, they show a rather loose organization that sometimes does not fit the red line of the results section. For example, Figure 1D is not mentioned in the paragraph that refers to several other panels of the same figure (see lines110-128). Panel 1D is mentioned later in the discussion. Does 1D really fit into Figure 1 then? Are all the panels indeed required to be shown in the main document? As some elements are only briefly mentioned, the authors might also consider moving some into the supplement (e.g. left part of Figure 1C, Figure 2A, Figure 3B ...) or at least try to distribute some panels into more figures. This would reduce complexity and increase comprehensibility for future readers. Also, Figure 3 is a way too complex. Panel G could be an alone-standing figure. The latter would also allow for an increase in font sizes or to show ChIP data of both conditions (L+O2 and D-O2) separately. Moreover, a figure legend typically introduces the content as a whole by one phrase but here only the different panels are described, which fits to the impression that all the different panels are not well connected. Of course, it is the decision of the authors what to present and how but may they consider restructuring and simplifying.

According to the advice, we have rearranged the Figure composition.

The left side of Figure 1C has been moved to supplement. Instead, representative expression fold changes of “Transient”, “Plateau”, “Continuous”, and “Late” genes are shown for comprehensibility. We left Figure 1D in Figure 1, as this diagram shows our motive to focus on hox and nifJ. We moved Figure 2A to supplement. We did not move Fig3B, as this figure shows the distribution of cyAbrB2 (“long tract of AT-rich DNA”) comprehensively and simply. We agree that Figure 3 was too complex. Therefore, we moved Figures 3F and 3G to a new independent figure (Figure 4). In Figure 4C (former 3G), we show the ChIP data of the L+O2 condition only, and the change of ChIP data under the D-O2 condition is shown in Figure 5. The schematic image showing cyanobacterial chromosome and NAPs (previous Figure 5E) was omitted because it was overinterpreting.

The authors assume a physiological significance of transient upregulation of e.g. hox genes under microoxic conditions. But does the hydrogenase indeed produce hydrogen under the conditions investigated and is this even required? Moreover, the authors use the term "fermentative gene". But is hydrogen indeed a fermentation product, i.e. are protons the terminal electron acceptor to achieve catabolic electron balance? Then huge amounts of hydrogen should be released. Comment should be made on this.This is a very important point; Yes, hydrogenase indeed produces hydrogen under the conditions we investigated, and proton accepts a majority of reducing power under the dark microoxic condition. We wrote in the introduction section as follows; “*Hydrogen is generated in quantities comparable to lactate and dicarboxylic acids as the result of electron acceptance in the dark microoxic condition* (Akiyama and Osanai 2023; Iijima et al. 2016)” (ll.54-55). The detailed explanation is below, although omitted from the manuscript.

A recent study (Akiyama and Oasanai 2023) quantified the consumed glycogen and secreted fermentative products (hydrogen, lactate, dicarboxylic acid, and acetate) in the *Synechocystis* under the dark microoxic condition, the same conditions as we investigated. The system of the study consists of a 10 mL liquid layer and a 10 mL gas layer, cultivated for 3 days under dark microoxic conditions. Then the amounts of lactic acid, dicarboxylic acid, and hydrogen were approximately 2 µmol, 3.5 µmol, and 11µmol (assuming the gas layer was at 1 atm and ignoring aqueous population), respectively. On the other hand, glycogen equivalent to 15µmol of glucose was consumed in the system. This estimate supports hydrogen accounts for a substantial portion of fermentative products during dark microoxic conditions.

The necessity of hydrogen production under dark microoxic conditions was demonstrated in (Gutekunst et al. 2014). They show hydrogenase activity is required for the mixotrophic growth in the light-dark and microoxic cycle with arginine. The necessity remains unclear in our conditions because we only performed continuous dark microoxic conditions without glucose.

The authors also mention a reverse TCA cycle. But is its existence an assumption or indeed active in cyanobacteria, i.e. is it experimentally proven? The authors are a little bit vague in this regard (see lines 241-246).

We misused the Terminology. We mean to mention the “reductive branch of TCA”. Cyanobacteria conduct the branched TCA cycle under microoxic conditions. One of the branches is the reductive branch, which reduces oxaloacetate to produce malate. We corrected “reverse TCA cycle” to “reductive branch of TCA”. (Figure 1D and ll.260-262)

**Reviewer #2 (Public Review):**
This work probes the control of the hox operon in the cyanobacterium Synechocystis, where this operon directs the synthesis of a bidirectional hydrogenase that functions to produce hydrogen. In assessing the control of the hox system, the authors focused on the relative contributions of cyAbrB2, alongside SigE (and to a lesser extent, SigA and cyAbrB1) under both aerobic and microoxic conditions. In mapping the binding sites of these different proteins, they discovered that cyAbrB2 bound many sites throughout the chromosome repressed many of its target genes, and preferentially bound regions that were (relatively) rich in AT-residues. These characteristics led the authors to consider that cyAbrB2 may function as a nucleoid-associated protein (NAP) in Synechocystis, given its functional similarities with other NAPs like H-NS. They assessed the local chromosome conformation in both wild-type and cyabrB2 mutant strains at multiple sites within a 40 kb window on either side of the hox locus, using a region within the hox operon as bait. They concluded that cyAbrB2 functions as a nucleoid-associated protein that influences the activity of SigE through its modulation of chromosome architecture.The authors approached their experiments carefully, and the data were generally very clearly presented and described.Based on the data presented, the authors make a strong case for cyAbrB2 as a nucleoid-associated protein, given the multiple ways in which it seems to function similarly to the well-studied *Escherichia coli* H-NS protein. It would be helpful to provide some additional commentary within the discussion around the similarities and differences of cyAbrB2 to other nucleoid-associated proteins, and possible mechanisms of cyAbrB2 control (post-translational modification; protein-protein interactions; etc.). The manuscript would also be strengthened with the inclusion of biochemical experiments probing the binding of cyAbrB2, particularly focusing on its oligomerization and DNA polymerization/bridging potential.

We agree with the comment that the biochemical experiments will deepen our insights into the cyAbrB2 and chromatin conformation. As the reviewer pointed out, the biochemical assay will provide valuable information on mechanisms of cyAbrB2 control, such as post-transcriptional modification, cooperation with cyAbrB1, oligomerization, and the structure of cyAbrB2-bound DNA. However, we think those potential findings are worth of new independent research paper, rather than a part of this paper. Therefore, we added a discussion mentioning biochemistry as the future work (ll.275-290; the section of “The biochemistry of cyAbrB2 will shed light on the regulation of chromatin conformation in the future”).

Previous work had revealed a role for SigE in the control of hox cluster expression, which nicely justified its inclusion (and focus) in this study. However, the results of the SigA studies here suggested that SigA both strongly associated with the hox promoter, and its binding sites were shared more frequently than SigE with cyAbrB2. The focus on cyAbrB2 is also well-justified, given previous reports of its control of hox expression; however, it shares binding sites with an essential homologue cyAbrB1. Interestingly, while the B1 protein appears to bind similar sites, instead of repressing hox expression, it is known as an activator of this operon. It seems important to consider how cyAbrB1 activity might influence the results described here.

We infer that the minor side of the bimodal SigE peak is the genuine population that contributes to hox transcription, as hox genes are expressed in a SigE-dependent manner (Figure S2). We considered the strong SigA peak upstream of the hox operon binds the promoter of TU1715, the opposite direction of the hox operon. We added a description of the single SigA peak and bimodal SigE peak near the TSS of the hox operon as follows;

“A bimodal peak of SigE was observed at the TSS of the hox operon in a microoxic-specific manner (Figure 6C bottom panel). The downstream side of the bimodal SigE peak coincides with SigA peak and the TSS of TU1715. Another side of the bimodal peak lacked SigA binding and was located at the TSS of the hox operon (marked with an arrow in Figure 6C), although the peak caller failed to recognize it as a peak.” (ll.206-209)

The point that cyAbrB1 binds similar sites as cyAbrB2, despite regulating hox expression in the opposite direction, is very interesting. Therefore, we referred to the transcriptome data of the cyAbrB1 knockdown strain and compared the impact of cyAbrB1 knockdown and cyAbrB2 deletion. We described in result and discussion as follows;

“we referred to the recent study performing transcriptome of cyAbrB1 knockdown strain, whose cyAbrB1 protein amount drops by half (Hishida et al. 2024). Among 24 genes induced by cyAbrB1 knockdown, 12 genes are differentially downregulated genes in cyabrb2∆ in our study (Figure S5D).” (ll.162-165)

“CyAbrB1, the homolog of cyAbrB2, may cooperatively work, as cyAbrB1 directly interacts with cyAbrB2 (Yamauchi et al. 2011), their distribution is similar, and they partially share their target genes for suppression (Figures 3A S5C and S5D). The possibility of cooperation would be examined by the electrophoretic mobility shift assay of cyAbrB1 and cyAbrB2 as a complex. Despite their similar repressive function, cyAbrB1 and cyAbrB2 regulate hox expression in the opposite directions, and their mechanism remains elusive.” (ll.292-296)

*Hox* operon differs from this general tendency. To see if cyAbrB1 behaves differently from cyAbrB2 in the *hox* operon, we did an additional ChIP-qPCR experiment on cyAbrB1 in the aerobic condition and the dark microoxic condition (Figure 5C). However, we could not find the difference.

**Reviewer #1 (Recommendations For The Authors):**
Figure 1B: I recommend changing the header in the grey bar to terms like "upregulated" and "downregulated", which are also used in the legend description. Upregulation of genes can also be a result of de-repression, which is why the term "activated" is somewhat misleading.

Corrected.

Lines 114-116: It is unclear what the authors exactly mean here. Please clarify.

We rephrase the sentence “The enrichment in the butanoate metabolism pathway indicates the upregulation of genes involved in carbohydrate metabolism. We further classified genes according to their expression dynamics.” (ll.110-111)

**Reviewer #3 (Recommendations For The Authors):**
Major/experimental comments:(1) For the chromosome conformation capture experiments, it is indicated that these were conducted at aerobic (1hr) and microoxic (4 hr) conditions. But the data presented in Figure 1 suggest that 1 hr corresponds to the beginning of microoxic growth, and that time 0 is aerobic. The composite 3C data in Figure 5 show some interesting but specific differences. It is appreciated that the authors presented the profiles for individual samples in Figure S7, and the differences here do not seem to be as compelling. Are the major differences being highlighted significantly (statistically) different (e.g. at the (c) and (j) loci)? Might the differences be starker if an earlier aerobic condition (e.g. time 0) had been used instead of the 1 hr - microoxic - timepoint?

Previous Figure 5 consisted of three time points (solid line: aerobic condition, dashed line:1hr of microoxic condition, and dotty line:4hr of microoxic condition). We omitted data of 4hr in the main figure (Figure 7) as 4hr in microoxic conditions makes data complicated. Three time points are shown in the profiles of individual loci (Figure S8).

There is no statistical significance found in (c) and (j) loci by t-test. Therefore, we have toned down the interpretation of 3C data as follows; “Our 3C result demonstrated that cyAbrB2 influences the chromosomal conformation of hox and nifJ region to some extent (Figure 7).” (ll.325-326)

(2) This is a complicated system that involves multiple regulatory proteins, each of which is differentially affected by the growth conditions (aerobic/microoxic). It is obviously beyond the scope of this work to probe deeply into all of these proteins. The focus here was on cyAbrB2, and to a slightly lesser extent SigE; however, based on the data presented, it seems that SigA and cyAbrB1 may be equally important contributors to hox control/expression, and in the case of cyAbrB1, possibly also to chromosome conformation. cyAbrB1 appears to have the same binding sites as cyAbrB2, and has been reported to interact with cyAbrB2. Given this association, it is possible that the two proteins may affect the binding of each other, and that loss of one might lead to enhanced binding by the other (or binding may require heterooligomerization?). Probing the regulatory interplay between these two proteins (or at least discussing it) feels important. Conducting e.g. mobility shift assays with each protein, both individually and together, could possibly allow for some understanding of how they function together.

We agree that the biochemistry of cyAbrB2 and cyAbrB1 may explain why cyAbrB1 and cyAbrB2 bind long tracts of AT-rich genome regions in vitro. We would like to put the biochemistry future plan as we think biochemistry data is beyond the present study.

The idea that cyAbrB1 and cyAbrB2 cooperate to form heterooligomers and broad binding to the genome is a very rational and interesting prediction. We add this idea to the discussion “Overall, the biochemistry integrating assay conditions (PTM, buffer condition, and cooperation with cyAbrB1) and output (DNA binding, oligomerization, and DNA structure) will deepen the understanding of cyAbrB2 as cyanobacterial NAPs.”(ll.287-290). We also compared our transcriptome of ∆_cyabrb2 with the recent study of cyabrb1 knockdown (ll. 162-165), and concluded “they partially share their target genes for suppression (Figures 3A S5C and S5D)” (l. 293).

(3) Throughout the manuscript, there is reference made to cyAbrB2 binding becoming 'blurry' or non-specific under microoxic conditions. It is not clear what this means. It appears that when cyAbrB2 binds, any given protected region can be quite extensive, which can be suggestive of polymerization along the chromosome. Are the boundaries for binding sites typically clearly delineated, and this changes when the cultures are growing under microoxic conditions? There is also no mention made anywhere about oligomerization potential for cyAbrB2, which would be important for the polymerization, and bridging suggested for cyAbrB2 in the model presented in Figure 5. Previous publications (Song et al., 2022; Ishi et al., 2008) have suggested that it can exist as a dimer in vivo, but that in vitro it is largely monomeric. The manuscript would benefit from some additional biochemical analyses of cyAbrB2 binding activity, with a particular focus on DNA binding and oligomerization/bridging potential, and some additional discussion about these characteristics as well.Throughout the manuscript, there is reference made to cyAbrB2 binding becoming 'blurry' or non-specific under microoxic conditions. It is not clear what this means.

In order to clearly describe “cyAbrB2 binding becomes blurry”, we rearranged the figure composition and made an exclusive figure (Figure 5). We also rephrased the description by adopting the reviewer’s word “boundaries for binding sites”, as this phrase well describes the change. “When cells entered microoxic conditions, the boundaries of the cyAbrB2 binding region and cyAbrB2-free region became obscure (Figure 5), “(ll.319-320)

There is also no mention made anywhere about oligomerization potential for cyAbrB2,

We added the discussion about oligomerization “DNA-bound cyAbrB2 is expected to oligomerize, based on the long tract of cyAbrB2 binding region in our ChIP-seq data. However, no biochemical data mentioned the DNA deforming function or oligomerization of cyAbrB2 in the previous studies and preference for AT-rich DNA is not fully demonstrated in vitro (Dutheil et al. 2012; Ishii and Hihara 2008; Song et al. 2022)”(ll. 277-280) and “Overall, the biochemistry integrating assay conditions (PTM, buffer condition, and cooperation with cyAbrB1) and output (DNA binding, oligomerization, and DNA structure) will deepen the understanding of cyAbrB2 as cyanobacterial NAPs.” (ll.287-290)

The manuscript would benefit from some additional biochemical analyses of cyAbrB2 binding activity, with a particular focus on DNA binding and oligomerization/bridging potential, and some additional discussion about these characteristics as well.

We added the discussion integrally considering known features of cyAbrB2, novel findings on cyAbrB2, and the comparison with known NAPs (ll.269-290).

(4) Given that the major take-away for the authors (based on the title) seems to be the nucleoid-associated protein potential for cyAbrB2, the Discussion would benefit from some additional focus in this area. How similar is cyAbrB2 to other nucleoid-associated proteins? (e.g. H-NS, Lsr2) How does counter-silencing work for other nucleoid-associated proteins? Can the authors definitively exclude the possibility of binding site competition/occlusion, given that cyAbrB2 covers the promoter region of hox? What is other nucleoid-associated proteins have been characterized in the cyanobacteria?

We agree with the point, so we additionally discussed cyAbrB2 comparing with H-NS and Lsr2, the canonical NAPs (ll. 269-290).

We did not deny the possibility of the exclusion of RNAP by cyAbrB2, but the previous manuscript insufficiently discussed that. To emphasize that cyAbrB2 excludes RNA polymerase, we simplified Figure 6 and employed mosaic plots showing anti-co-occurrence of cyAbrB2 binding regions and SigE peaks. Furthermore, we added discussion about SigE exclusion by cyAbrB2 (ll. 355-359)

We mention the possibility of other nucleoid-associated proteins in cyanobacteria in the discussion. “Furthermore, the conformational changes by deletion of cyAbrB2 were limited, suggesting there are potential NAPs in cyanobacteria yet to be characterized.” (ll.336-339)

(5) Previous work (Song et al., 2022) showed that changing the AT content of cyAbrB2 binding sites did not affect its ability to bind DNA. There are also previous papers suggesting that cyAbrB2 may be subject to diverse post-translational modifications (e.g. phosphorylation - Spat et al., 2023; glutationylation - Sakr et al., 2013), as well as association with cyAbrB1. These collectively suggest there may be other factors that contribute to cyAbrB2 binding specificity/activity. These seem like relevant points to discuss, particularly given the transient nature of the cyAbrB2 effects on some genes.

We have included the discussion about AT content, post-translational modifications and transient regulations, and association with cyAbrB1 (ll. 284-295)

(6) Given the major binding site for SigA upstream of the hox operon, it seems that it likely also contributes to hox cluster expression, together with SigE. Is there a sense for the relative contribution of each sigma factor to hox cluster expression? And whether both are subject to the same inhibitory effect of cyAbrB2?

As described above response to the public review, the SigA binding site upstream of the hox operon should be assigned to the TSS of TU1715 (Figure 6C). Transcription of hox operon is highly dependent on SigE as shown in Figure S2, and residual transcription in *sigE*∆ strain is derived from other sigma factors (SigABCD). Estimating the relative contribution of sigma factors other than SigE is difficult at present because SigABCDE can partially compensate for each other.

As the different impact of NAPs on the primary and alternative sigma factor is observed in H-NS (Shin et al. 2005), whether both the primary sigma factor (SigA) and the alternative sigma factor (SigE) are inhibited by cyAbrB2 to the same extent is a very interesting question.

We calculated the odds ratio of SigE and SigA being in the cyAbrB2-free region and wrote in the result; “SigE preferred the cyAbrB2-free region in the aerobic condition more than SigA did (Odds ratios of SigE and SigA being in the cyAbrB2-free region were 4.88 and 2.74, respectively).” (ll.193-195) and discussed “The higher exclusion pressure of cyAbrB2 on SigE may contribute to sharpening the transcriptional response of hox and nifJ on entry to microoxic conditions.” (ll.357-359)

(7) The 3C experiments suggest there are indeed changes in chromosome architecture in the hox region as growth conditions change and when different regulators are present. Across the chromosome, analogous changes are expected; however, it may be premature to draw this conclusion based on changes at one locus. Is there a reason that the authors did not take full advantage of their 3C samples and sequence them, to capture the full chromosome interactome at the two time-points? This would allow broader conclusions to be drawn regarding changes in chromosome structure and the impact of cyAbrB2.

In response to the suggestion, we performed an additional 3C assay on the nifJ region by utilizing residual 3C samples. Expanding to genome-wide sequence (Hi-C) needs concentration of ligated fragments by the biotinylation, which were omitted in our 3C sample.

We rewrote the result as obtained from the 3C data of hox and nifJ (ll.220-245) and omitted the schematic image of an entire chromosome of cyanobacteria (previous Figure 5E).

Editorial comments:(1) The data presentation in Figure 1 is very effective.(2) Line 87: please rephrase - you can have 'high similarity' or 'high levels of identity', but not high levels of homology - genes/proteins are either homologous or not.(3) Line 118: classified into four 'groups'?(4) Line 590: remove 'the'.(5) Figure 2S, panel B: please define acronyms in the legend (GT, IP) and write out 'FLAG' in full for AbrB1.

(2) to (5) have been corrected.

(6) Please provide information on or a reference for the tagging of SigA for use in the ChIP-seq experiments within the Materials and Methods.

Added (l.365)

(7) Line 648: space between 'binding' and 'regions'.

corrected.

(8) Fig 4E: please make the solid lines thicker - they are currently difficult to see.

We have made Figure 6C (former 4E) larger and the line thicker.

(9) Line 666: location.(10) Line 673: Individual.(11) Figure S5, panel C graph title: should this be 'Relative'?(12) Figure S7: What is 'GT'? Should this be 'WT'?

(9) to (12) have been corrected.

(13) In addition to the data presented in Figure 3G, it would be nice to have a small table or Venn diagram summarizing the number of cyAbrB2 binding sites that fall into the different categories (full gene/operon; downstream of a gene; within a gene; promoter region).

In response to the comment, we noticed the categories we had applied (full gene/operon; downstream of a gene; within a gene; promoter region) were arbitrary. Therefore, we categorized transcriptional units (TUs) according to the extent of occupancy by cyAbrB2. (Figures 4B and 4C)

(14) Line 280-281: suggest replacing 'mediates' with 'influences'. 'Mediates' sounds like a direct interaction (for which the evidence is not currently strong without some additional biochemical data), but 'influences' could better accommodate both direct and indirect possibilities.(15) Line 410: it is not clear what this means.

We have omitted “As a result, DNA ~600-fold condensed DNA than 3C samples were ligated.”, as it does not give any information about the experimental procedure.